# Dynamic nanoscale architecture of synaptic vesicle fusion in mouse hippocampal neurons

Jana Kroll [1,2,3] ✉, Uljana Kravčenko[4,5], Mohsen Sadeghi [6,7], Christoph A. Diebolder [8,9], Lia Ivanov[1], Małgorzata Lubas[1], Thiemo Sprink[8,9], Magdalena Schacherl [10], Mikhail Kudryashev [4,10] & Christian Rosenmund [1] ✉

Synaptic vesicle (SV) fusion is not only tightly coordinated but also happens at a millisecond timescale. Competing models for fusion initiation and propagation suggest tight docking and hemifusion of SVs or localized lipid rearrangements leading to tip-like membrane contacts. Yet, a direct nanoscale examination of the full SV fusion sequence has been lacking. Here, we establish a workflow for timed in situ cryo-electron tomography of optogenetically stimulated mouse neurons to capture the complete SV fusion sequence – from SV recruitment to fusion pore formation, opening and collapse – with near-native structural preservation. Notably, tethered SVs directly undergo fusion initiation via stalk formation, without preceding tight docking or SV flattening. The plasma membrane forms a minimal dimple during fusion initiation, contradicting preceding models that invoke strong membrane bending prior to fusion. In addition, we observe filaments linking fusing SVs to adjacent SVs, indicating a physical link between fusion and SV resupply.

Neurotransmitter release is driven by the rapid fusion of synaptic vesicles (SVs) with the presynaptic plasma membrane, a process fundamental to neuronal communication. The presynaptic active zone (AZ) harbors release sites, where SVs are coupled to voltage-gated calcium channels and primed for fusion[1,2]. The synaptic fusion machinery, consisting of soluble *N*-ethylmaleimide-sensitive factor attachment protein receptors (SNAREs), regulatory proteins like Munc13 and Munc18, and the calcium sensor synaptotagmin-1, catalyzes the fusion reaction, helping to overcome energy barriers imposed by repulsive forces during SV-AZ membrane apposition and lipid reorganization[1,3,4]. An SV has reached a primed state when trans-SNARE complexes have formed and closely interact with synaptotagmin-1 and complexin[5,6]. Calcium influx triggers membrane binding of synaptotagmin-1, initiating the vesicle fusion process within a fraction of a millisecond[7–9].

While much is known about the molecular players of the fusion apparatus, the precise structural pathway by which SV and plasma membranes fuse remains largely unresolved. Advances in electron

[1]Charité – Universitätsmedizin Berlin, corporate member of Freie Universität Berlin and Humboldt-Universität zu Berlin, Institute of Neurophysiology, Berlin, Germany. [2]Max Delbrück Center for Molecular Medicine in the Helmholtz Association, Structural Biology of Membrane-Associated Processes, Berlin, Germany. [3]Freie Universität Berlin, Institute of Chemistry and Biochemistry, Berlin, Germany. [4]Max Delbrück Center for Molecular Medicine in the Helmholtz Association, In Situ Structural Biology, Berlin, Germany. [5]Humboldt-Universität zu Berlin, Department of Biology, Berlin, Germany. [6]Freie Universität Berlin, Department of Mathematics and Computer Science, Berlin, Germany. [7]Leibniz-Forschungsinstitut für Molekulare Pharmakologie, Berlin, Germany. [8]Max Delbrück Center for Molecular Medicine in the Helmholtz Association, Technology Platform Cryo-EM, Berlin, Germany. [9]Charité – Universitätsmedizin Berlin, corporate member of Freie Universität Berlin and Humboldt-Universität zu Berlin, Core Facility for Cryo-Electron Microscopy, Berlin, Germany. [10]Charité – Universitätsmedizin Berlin, corporate member of Freie Universität Berlin and Humboldt-Universität zu Berlin, Institute of Medical Physics and Biophysics, Berlin, Germany. ✉e-mail: jana.kroll@mdc-berlin.de; christian.rosenmund@charite.de

microscopy (EM) sample preparation techniques, combining optogenetic or electrical stimulation and high-pressure freezing, have provided snapshots of SV fusion at central synapses. Membrane pits thought to represent fusion events were observed several milliseconds after stimulation[10], along with a decreased number of morphologically docked SVs[10–13]. However, the required dehydration, freeze-substitution and resin embedding of EM samples may introduce artifacts that impede the direct observation of native membrane dynamics[14]. In contrast, in situ cryo-electron tomography (cryo-ET) provides a near-native sample preservation, going along with a higher structurally interpretable resolution[15–19]. This led to differences in the assessment of SV-plasma membrane docking activity[17]. In addition, cryo-ET enabled the characterization of proteinaceous filamentous tethers that may contribute to the functional state of SVs[15,20–23]. Yet, adding millisecond (ms) temporal resolution has so far been challenging[24]. Although a small number of SV fusion intermediates have been captured using a spraying technique for synaptic stimulation recently[22], a systematic characterization of SV fusion in situ is still lacking. Our current understanding of membrane dynamics during SV fusion is therefore essentially built on in vitro[25–27] and in silico[8,28–31] studies of membrane fusion events.

Several mechanistic models have been proposed to explain how SNARE zippering and synaptotagmin-1 membrane binding catalyze membrane contact and merger of SV and plasma membranes. Previous studies utilizing elastic continuum models suggested that SV fusion proceeds through energetically metastable hemifusion diaphragms - intermediates where the outer leaflets of the membranes merge while inner leaflets remain distinct[32,33]. According to simulations, this hemifusion state could be reached through a point-like contact formed between the two membranes via bending of their lipid bilayers[32,33]. Other models suggest that SV and plasma membranes can form large close-contact interfaces, with fusion nucleating at the edge of the contact zone through SNARE zippering[3,25,26].

Alternative models, particularly from coarse-grained simulations, suggest that SV fusion can initiate between flat bilayers via highly localized lipid rearrangements. These include acyl chain splaying events, which create a small yet rapidly expanding hydrophobic core, seeding the formation of a fusion stalk[34]. A recent all-atom simulation of SV fusion showed that lipid chain splaying, resulting in a hydrophobic bridge and fusion pore formation, can be initiated by calcium-triggered rearrangements of the SNARE complex[8]. Despite the advances, it remains unclear which of these models accurately reflects the SV fusion process at central synapses, primarily because a direct visualization of the complete SV fusion process has thus far not been possible.

In this study, we develop a workflow for timed in situ cryo-ET using optogenetics. We utilize this workflow for a comprehensive reexamination of SV fusion under near-physiological conditions and within the native cellular environment. Our analyses reveal that stalk formation leads to the formation of a symmetric fusion pore without tight docking or hemifusion. Furthermore, we uncover structural evidence for mechanically coordinated SV resupply, suggesting a link between SV fusion and replenishment.

## Results

### Coupled optogenetic stimulation and cryofixation of mouse hippocampal neurons

To characterize the synaptic nanoscale architecture during and shortly after neurotransmitter release, we developed a workflow combining optogenetics and in situ cryo-ET (Fig. 1). For optogenetic stimulation, we expressed the channelrhodopsin-2 (ChR2) variant ChR2(E123T/T159C) in murine hippocampal neurons, which was shown to induce action potentials in a particularly fast and robust manner[35,36]. In addition to the established ChR2(E123T/T159C) version, which includes YFP for cellular localization (Fig. 1b), we used additional constructs

harboring a Cerulean or mScarlet to avoid potential spectral overlap with fluorescent sensors in subsequent experiments (Supplementary Fig. 1a). We performed electrophysiological recordings of neurons infected with ChR2(E123T/T159C)-YFP or one of the two new constructs ChR2(E123T/T159C)-Cerulean and ChR2(E123T/T159C)-mScarlet to assess their efficacy to induce action potentials. Onset of a light pulse resulted reliably in an action potential with an average delay of 4.6-4.8 ms, a similar value as described for ChR2(E123T/T159C)-YFP in a similar experimental setting[37] (Supplementary Fig. 1b), and a robust response to light stimuli up to a frequency of 40 Hz for all three tested constructs (Supplementary Fig. 1c and 1d).

Neurons cultured on EM grids (for details of our cell culture setup, see Supplementary Fig. 2) were plunge frozen at DIV16-18 using a modified Vitrobot Mark IV (Thermo Fisher Scientific) plunge freezer equipped with an LED connected to optical fibers inside and below the chamber (Fig. 1b and Supplementary Fig. 3a, b). Optogenetic stimulation (2 pulses at 10 Hz) was performed at ~37 °C and an elevated extracellular calcium concentration of 4 mM to increase the vesicular release probability. The first stimulus was applied within the chamber at a maximum of 100 ms before vitrification for efficient fluorescent labeling of stimulated synapses (see Supplementary Fig. 4c). The second stimulus, which induced the fusion events morphometrically characterized in this study, was applied while the sample grid traveled towards the cooled ethane. This second stimulus started approximately 7 ms before vitrification, inducing an action potential 2–5 ms before the grid was dipped into cooled ethane (additional cooling time of the sample grid to 0 °C < 1 ms[38]). The exact timing of the LED pulses and freezing were monitored using a high-speed camera (Supplementary Fig. 3c). Considering that most action potentials were induced 3–6 ms after light onset in our electrophysiological experiments (see Source Data) and that the delay between action potential generation at the presynapse and synchronous neurotransmitter release is typically 1 ms or shorter[39], our setup was well suited for cryofixing neurons shortly before, during, and directly after neurotransmitter release.

### Confirmation of neurotransmitter release using the glutamate sensor iGluSnFR3 in cryofixed neurons

After plunge freezing, we aimed to validate successful stimulation using a fluorescent biosensor for synaptic activity. For this purpose, we first characterized the kinetics and fluorescence intensity changes of the calcium sensor, SynGCaMP6f[40], and different variants of the glutamate sensor, iGluSnFR, via live imaging of neurons cultured on coverslips (Fig. 2a and Supplementary Fig. 4a). Of all tested constructs, the fluorescent glutamate sensor iGluSnFR3.v857.GPI containing a GPI anchor for postsynaptic enrichment ([41], from now iGluSnFR3) yielded the best-fitting properties with a maximum fluorescence intensity of $0.4 \pm 0.03 \ \Delta F/F_0$, (Supplementary Fig. 4b), an increase to half-maximum of $\tau_{50\%} = 22.6 \pm 3$ ms (Supplementary Fig. 4d), and an increase to maximal intensity of $\tau_{max} = 64.7 \pm 6.2$ ms (Supplementary Fig. 4c). To examine the cellular localization of iGluSnFR3 signals, we performed a *post hoc* immunofluorescence staining of synaptic proteins on samples used for live imaging (Supplementary Fig. 4f). With this correlation, we could verify that action potential-induced iGluSnFR3 signals overlap primarily with the postsynaptic marker Homer1.

Assuming that the glutamate-bound, highly-fluorescent conformation of iGluSnFR3 can be preserved under cryogenic conditions, we acquired cryo-confocal stacks of optogenetically stimulated and plunge frozen neurons expressing iGluSnFR3 (Fig. 1c). We compared fluorescence intensities in neurites of unstimulated neurons, optogenetically stimulated neurons, and optogenetically stimulated and tetrodotoxin-treated (TTX, pharmacologically blocks sodium channels required for action potential induction) neurons of four independent cultures (two for TTX, Fig. 2b). In all four cultures, of which two were infected with ChR2(E123T/T159C)-YFP, one with ChR2(E123T/T159C)-

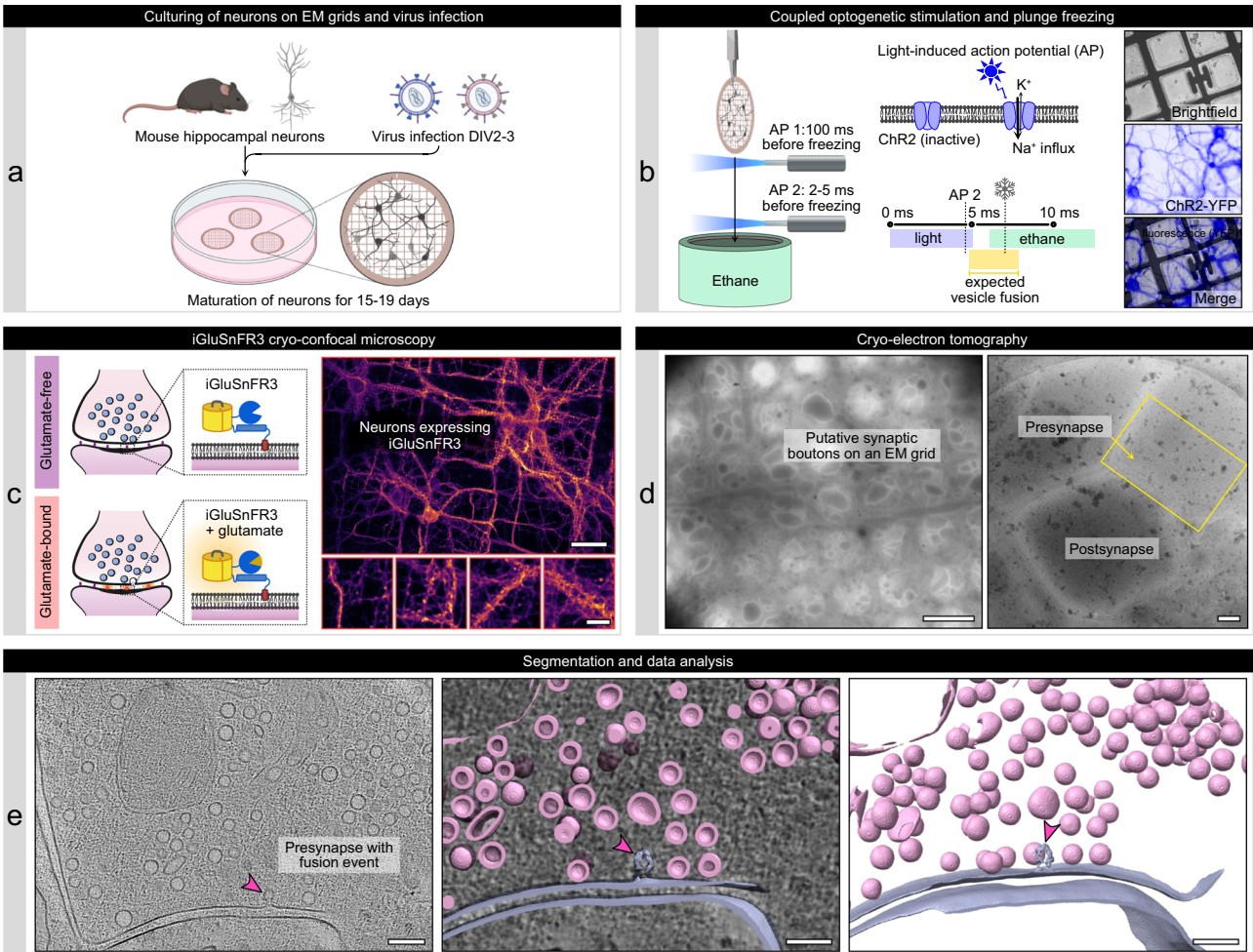

**Fig. 1 | Workflow combining optogenetic stimulation of neurons, iGluSnFR cryo-confocal microscopy, and in situ cryo-ET. a** Mouse hippocampal neurons are cultured on EM grids and infected with viruses for ChR2(ET/TC) and iGluSnFR3 expression. **b** Light pulses induce action potentials 100 ms (stim1) and 2-3 ms (stim2) before cryofixation of neurons via plunge freezing. Right panels: Cryo-fluorescence microscopy of ChR2(ET/TC)-YFP in plunge frozen neurons. **c** The upper right panel shows an overview of several neurons on an EM grid, scale bar 50 μm, the lower panels are zoom-ins to individual neurites containing synapses, scale bar 10 μm. **d** Cryo-ET tilt series were acquired from stimulated and control EM grids. Left: overview of a grid mesh with neurites and synaptic boutons, scale bar

5 μm. Right: Synapse within a hole of a holey carbon grid, scale bar 200 nm. **e** Tomograms are reconstructed from tilt series and used for segmentation and data analysis. The tomogram slice and segmentation show a stimulated synapse with ongoing SV fusion (pink arrowhead). Scale bars 200 nm. The reproducibility of the coupled optogenetic stimulation and plunge freezing was confirmed via cryo-confocal microscopy of iGluSnFR3 in 4 independent cell cultures/ plunge freezings, with in total 9 unstimulated EM grids, 13 stimulated EM grids, and 3 stimulated and TTX-treated EM grids. Schematic illustrations in panels (**a**, **b**) were created in BioRender. Kudryashev, M. (2025) https://BioRender.com/j5cdwwd and Kudrya-shev, M. (2025) https://BioRender.com/okc4479.

mScarlet, and one culture with both, the mean fluorescence intensity of the stimulated samples was consistently higher than of the unsti-mulated samples (for details, see Source Data), indicating that our setup combining optogenetic stimulation and plunge freezing works reliably.

We therefore pooled the measurements of the individual cultures and calculated fluorescence intensity histograms for all three conditions (Fig. 2c). In the stimulated samples, the mean fluorescence intensity of $36.3 \pm 1.3$ AU was significantly higher than in the two control conditions (unstimulated: $30.0 \pm 1.1$ AU, TTX-treated: $31.0 \pm 1.5$ AU, Kruskal-Wallis- test $p < 0.001$, KWS = 15.12, Fig. 2d). Likewise, the fraction of pixels with a high fluorescence intensity ($> 70$ AU), likely reflecting glutamate-bound iGluSnFR3, was significantly increased after stimulation ($20.6 \pm 0.9\%$ vs. $13.0 \pm 0.9\%$ and $11.9 \pm 1.2\%$ without stimulation and after TTX treatment, respectively, Kruskal-Wallis- test $p < 0.001$, KWS = 38.16, Fig. 2e). Of note, the difference in mean fluorescence intensities of optogenetically stimulated and control conditions was less pronounced under cryogenic conditions than during live imaging, likely because low-expressing neurons were not selected

during live imaging and non-responding neurons were excluded from the data analysis (no measurable response in 14%, weak response of $\Delta F/F_0 < 0.1$ in 23% of recordings for iGluSnFR3.v857.GPI, see Supplementary Methods and Source Data), whereas cryo-confocal stacks were acquired from regions selected blindly.

To test if high iGluSnFR3 fluorescence intensity can be used as a marker for SV fusion events, we performed correlative cryo-confocal microscopy and cryo-EM (cryo-CLEM) on a stimulated EM grid (Fig. 2f and Supplementary Fig. 5). As visible in the overview of four grid meshes, the overall morphology of neurons was preserved in our on-grid cell culture system. For tilt series acquisition, we selected regions containing a high density of neurites but no cell somata (yellow box in the middle panel of Fig. 2f). Correlating fluorescence and transmission electron microscopy (TEM), we observed the highest fluorescence intensity around large boutons likely resembling synapses (yellow arrowheads and box in Supplementary Fig. 5). The correlation of iGluSnFR3 fluorescence and tomogram slice (Fig. 2f') revealed that the highest fluorescence intensity was visible at the synaptic cleft between a presynapse and a postsynaptic

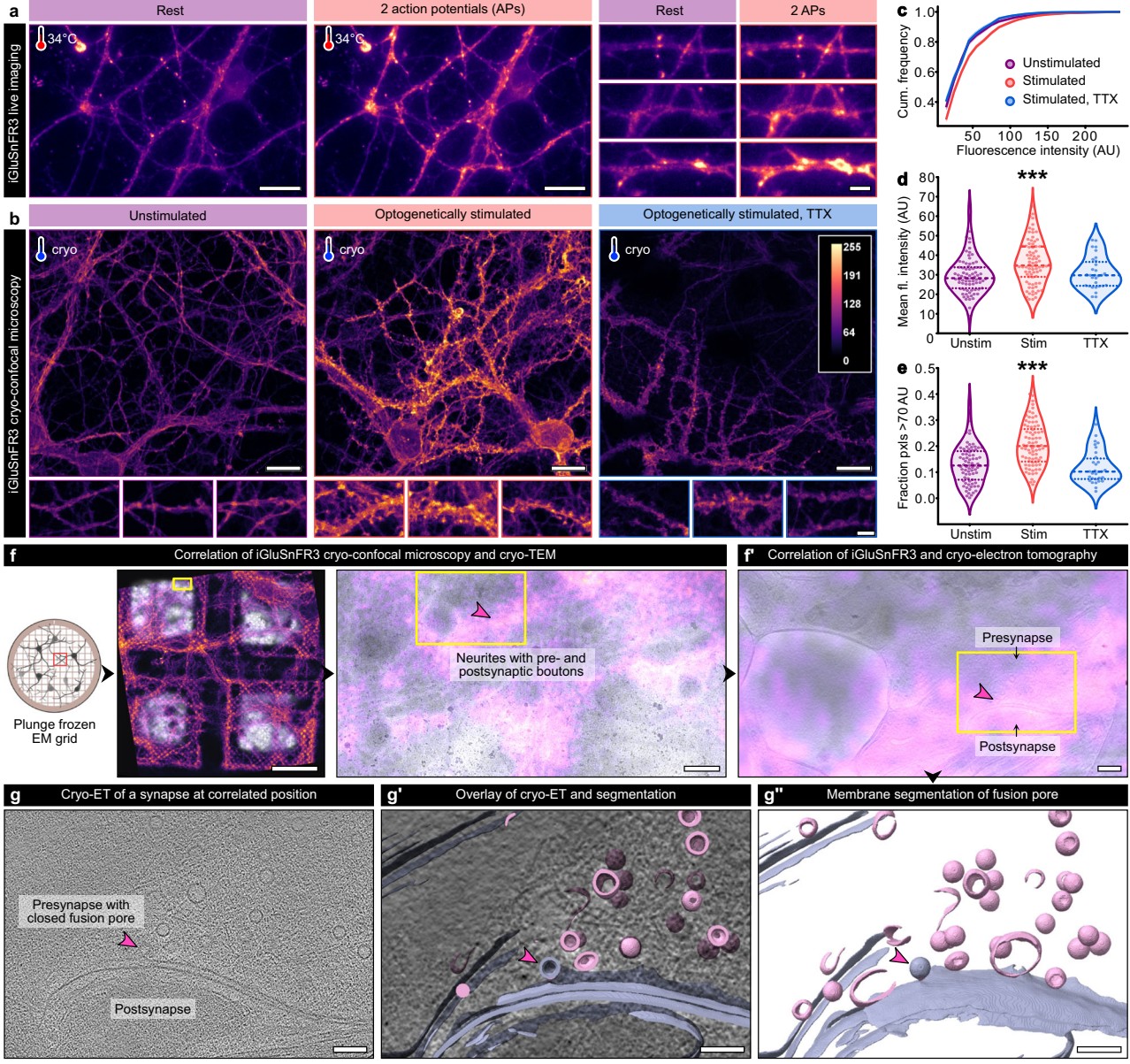

**Fig. 2 | Confirmation of synaptic glutamate release in stimulated, cryofixed neurons. a** Electrical field stimulation and live imaging of iGluSnFR3 at near-physiological temperature. Left panels: before stimulation, right panels: after stimulation. Scale bars: overviews 20 μm, zoom-ins 10 μm. **b** Maximum intensity projections of cryo-confocal stacks from unstimulated and optogenetically stimulated hippocampal neurons without and with TTX treatment. Scale bars upper panels: 20 μm, lower panels: 5 μm. **c–e** Cumulated fluorescence intensity histograms (**c**), mean fluorescence intensity (**d**), and fractions of pixels with a high fluorescence intensity > 70 AU (**e**) measured in areas containing individual neurites. Unstimulated: *N* = 77 confocal stacks from 9 grids and 4 independent cultures; stimulated: *N* = 80 stacks from 13 grids and 4 cultures; stimulated TTX:

*N* = 30 stacks from 3 grids and 2 cultures. Dashed lines in the violin plots indicate the median, dotted lines the 25% and 75% percentile. **d** Kruskal-Wallis test, *p* < 0.001, KWS = 15.12. **e** Kruskal-Wallis test, *p* < 0.001, KWS = 38.16. **f** Correlative iGluSnFR3 cryo-confocal microscopy and cryo-TEM of a stimulated grid. Left panel: overview of four grid meshes, scale bar 50 μm. Right panel: zoom into a region containing synaptic boutons, scale bar 500 nm. **f'** Correlation of iGluSnFR3 fluorescence and a reconstructed tomogram slice of a synapse, scale bar 200 nm. **g–g''** Tomogram slice (**g**), overlay (**g'**), and segmentation (**g''**) of a putative closed fusion pore (pink arrowhead) from the correlated synapse in (**f**). Scale bars 100 nm. The schematic illustration in panel (**f**) was created in BioRender. Kudryashev, M. (2025) https://BioRender.com/j5cdwwd. Source data are provided as a Source Data file.

bouton. At the presynaptic AZ membrane, a putative forming fusion pore was observed (Fig. 2g).

### In situ cryo-ET of SV fusion intermediates

Having confirmed that neurons cultured on EM grids were optogenetically stimulated before being subsequently plunge frozen, we acquired cryo-ET data from those grids to morphometrically and biophysically characterize SV fusion (Fig. 1d, e). Although we could show that the iGluSnFR3 fluorescence signal is per se suited to select regions of interest for cryo-ET, we acquired tilt series without

correlation of each position to avoid ice contaminations, which form during prolonged cryo-confocal microscopy, and to avoid the thinning of our samples via focused ion beam (FIB)-milling, going along with a lower sample throughput. We therefore acquired tilt series from regions with good cell and ice quality after confirming successful stimulation via cryo-fluorescence microscopy at only a few positions of the grids. From the acquired tilt series, we reconstructed 312 tomograms from stimulated and 95 tomograms from control (TTX-treated) samples (see Supplementary Video 1 for an exemplary tomogram of a stimulated synapse). We screened each tomogram manually for

synapses with a visible, cross-sectioned AZ. This resulted in 75 synapses in the stimulated and 28 synapses in the TTX-treated samples that were used for further analysis. Each AZ was then examined in more detail for membrane rearrangements that may be attributed to SV fusion (see Supplementary Fig. 6 for examples of excluded structures). Based on all observed events, and in accordance with previously described SV fusion intermediates[3,21,22], we defined seven categories: invagination of the AZ membrane (1), stalk formation (2), closed fusion pore (3), open fusion pore (4), dilating fusion pore (5), collapsing fusion pore (6), and small bumps (7) (Fig. 3a, b, Supplementary Fig. 8, see Supplementary Fig. 7 for morphometric criteria of each category). In addition to exemplary tomogram slices (Fig. 3b), we visualized 3D volumes of each category using UCSF ChimeraX[42] (Fig. 3c). We further performed subtomogram averaging (StA) of selected fusion events from most categories using Dynamo[43] (Supplementary Fig. 8a) and applied C61 symmetry (Supplementary Fig. 8b) to visualize the general membrane shape and bending. We were not able to generate a StA of open fusion pores because this category was particularly heterogeneous with open pore widths ranging from 2 to 18 nm.

SVs of category 1 (n = 10 SVs in stimulated synapses) were spherical, and in a distance of $6.4 \pm 0.4$ nm to the AZ membrane (Fig. 3a), the AZ membrane below the center of the SV was slightly invaginated ($2.6 \pm 0.3$ nm; Supplementary Fig. 8c and Source Data). In category 2 (stalk formation, n = 22), SVs were droplet-shaped with an evagination at the SV bottom, while the AZ membrane was almost flat or slightly invaginated. At closed fusion pores belonging to category 3 (n = 6), the SV and AZ membrane had already fused, resulting in a continuous membrane, as confirmed by grayscale intensity measurements along the stalk/fusion pore neck (Supplementary Fig. 7 and Source Data). In comparison to category 2, closed fusion pores were slightly taller, the invagination of the AZ membrane was more pronounced. Open fusion pores (category 4, n = 7) were smaller than closed fusion pores, the width of the open pores was between 2.3 and 18.3 nm. Based on our definition, open pores contained outward (positive) and inward (negative) membrane curvature at the pore neck, whereas dilating pores (category 5, n = 8) only showed inward curvature at the junction of SV and cell membrane, as illustrated in Supplementary Fig. 7. The side walls of dilating pores were vertical or angular. Compared to dilating pores, collapsing fusion pores (category 6, n = 12) were lower (dilating pore: $48.4 \pm 2.6$ nm vs. collapsing pore: $29.6 \pm 2.1$ nm) and wider (dilating pore: $37.4 \pm 1.1$ nm vs. collapsing pore: $50.4 \pm 2.3$ nm). Interestingly, the top membrane of collapsing fusion pores appeared thickened in relation to the surrounding AZ membrane by a factor of 1.5 (Fig. 3a and Supplementary Fig. 8c). Small bumps (category 7, n = 21) were again lower ($7.7 \pm 0.4$ nm) than collapsing fusion pores and varied in size and shape.

Overall, optogenetically stimulated synapses showed a significantly different abundance of membrane rearrangements compared to TTX-treated control synapses (Fisher's exact test, p = 0.007). We observed higher fractions of events in categories 1–4 and 6, with the most striking difference in category 2 vs. no membrane rearrangements (corrected Fisher's exact test, q = 0.035), in stimulated neurons (Fig. 3d), whereas the fraction of dilating pores (5) and small bumps (7) was alike under both conditions. Closed and open fusion pores were only present in the stimulated sample without TTX treatment. Therefore, we defined membrane rearrangements of categories 2–6 as ongoing SV fusion and cell membrane invaginations below tethered, round SVs (category 1) as events likely preceding SV fusion. Based on this definition, we observed ongoing SV fusion in 52% (39/75 synapses), bumps in 14.7% (11/75), and no membrane rearrangements in 33.3% (25/75) of all stimulated synapses. In the TTX-treated group, the distribution between these three conditions was significantly different (Fisher's exact test, p < 0.001): we found fusion events in 10.7% (3/28 synapses), bumps in 17.9% (5/28), and no membrane rearrangements in 71.4% (20/28) of all synapses (Fig. 3e).

Since some synapses contained more than one fusion event, we further counted each fusion event individually (Fig. 3f). Of all fusion events observed in the stimulated samples (N = 55), the majority was stalk formation (40%), followed by collapsing fusion pores (21.8%). Closed (10.9%), open (12.7%), and dilating (14.5%) fusion pores were less prevalent. Presuming that more transient and volatile conditions are stochastically less likely to be captured during plunge freezing, our observed numbers of events per fusion state may serve as a morphological readout for their speed. Based on this assumption, we generated a Markov state model to elucidate the kinetics of transitions between our defined states (Supplementary Fig. 9a). According to this model, fusing SVs may remain in state 2 (stalk formation) for comparatively longer, likely because energy barriers need to be overcome when the membranes of SV and AZ are approached and perturbed[3]. Alternatively or in addition, some of the formed stalks may not lead to SV fusion but instead get stuck or disassemble again[44,45]. We therefore made the transition from state 1 to state 2 reversible.

## Initiation of SV fusion via stalk formation

In previous studies, not only stalk formation but alternatively also (tight) docking has been suggested as prefusion state[3,26]. During tight docking, the SV approaches the AZ until the membranes are in direct and broad contact; the lipids of the SV and AZ membranes are supposed to intermix until they reach a hemifusion (diaphragm) state. In our examples of stalk formation, the SVs were droplet-shaped, and the average distance between SV and membrane was $4 \pm 0.3$ nm (smallest measured distance: 2.3 nm). This space between SV and AZ membrane appeared partially blurry in some of our examples (Supplementary Fig. 8c), which has previously been attributed to starting lipid intermixing[22]. In contrast, we observed one morphologically tightly docked SV at an AZ (Supplementary Fig. 8d), as well as one tightly docked SV (Supplementary Fig. 8e) and a putative hemifusion diaphragm (Supplementary Fig. 8f), both not in an AZ. To test whether SVs could still undergo tight docking in the early phase of SV fusion despite the high prevalence of stalk formation, we generated additional Markov state models assuming tight docking preceding or following stalk formation (Supplementary Fig. 9b, c, respectively). In both cases, mean waiting times exceeded the predefined total duration of 2 ms, indicating that these transitions are comparatively very unlikely. Together, our observations indicate that a transition from tethering to stalk and fusion pore formation without (tight) docking is likely the predominant fusion mechanism.

Furthermore, a slight invagination of the AZ membrane (state 1) was present below 6.5% of all tethered SVs within a distance of 4–8 nm from the AZ membrane (10/155 SVs). We observed these invaginations at stimulated synapses with and without additional fusion events. To test whether an invagination of the AZ membrane is beneficial for SV fusion initiation and may thus precede stalk formation, we generated a coarse-grained simulation of an SV approaching the AZ membrane (Fig. 3g). In this model, proteins (e.g., resembling synaptotagmin-1) actively induce membrane curvature as soon as the SV has reached a distance of 6 nm to the AZ and also interact with SNAREs. We thereby incorporated size and distance measurements of our morphometric analyses (Figs. 3a, 4c–g and Supplementary Fig. 11). With this model, we tested the effects of different concentrations of membrane curvature-inducing proteins on SV approximation and SNARE complex formation. Since previous studies reported on different copy numbers of synaptotagmin-1 per fusing SV, ranging from below $10^{46}$, approx. $15^{47}$ up to more than $20^{48}$, and also other curvature-inducing proteins may be present close to the fusion site, we tested for 0, 10, 20, 30, and 40 copy numbers of actively membrane curvature-inducing proteins. Interestingly, higher copy numbers of these proteins did not facilitate but rather impeded the recruitment of the SV, likely because SNARE complexes could not be formed efficiently anymore (Fig. 3h and Supplementary Fig. 10). Instead, 0 or 10 copies resulted in a fast SV

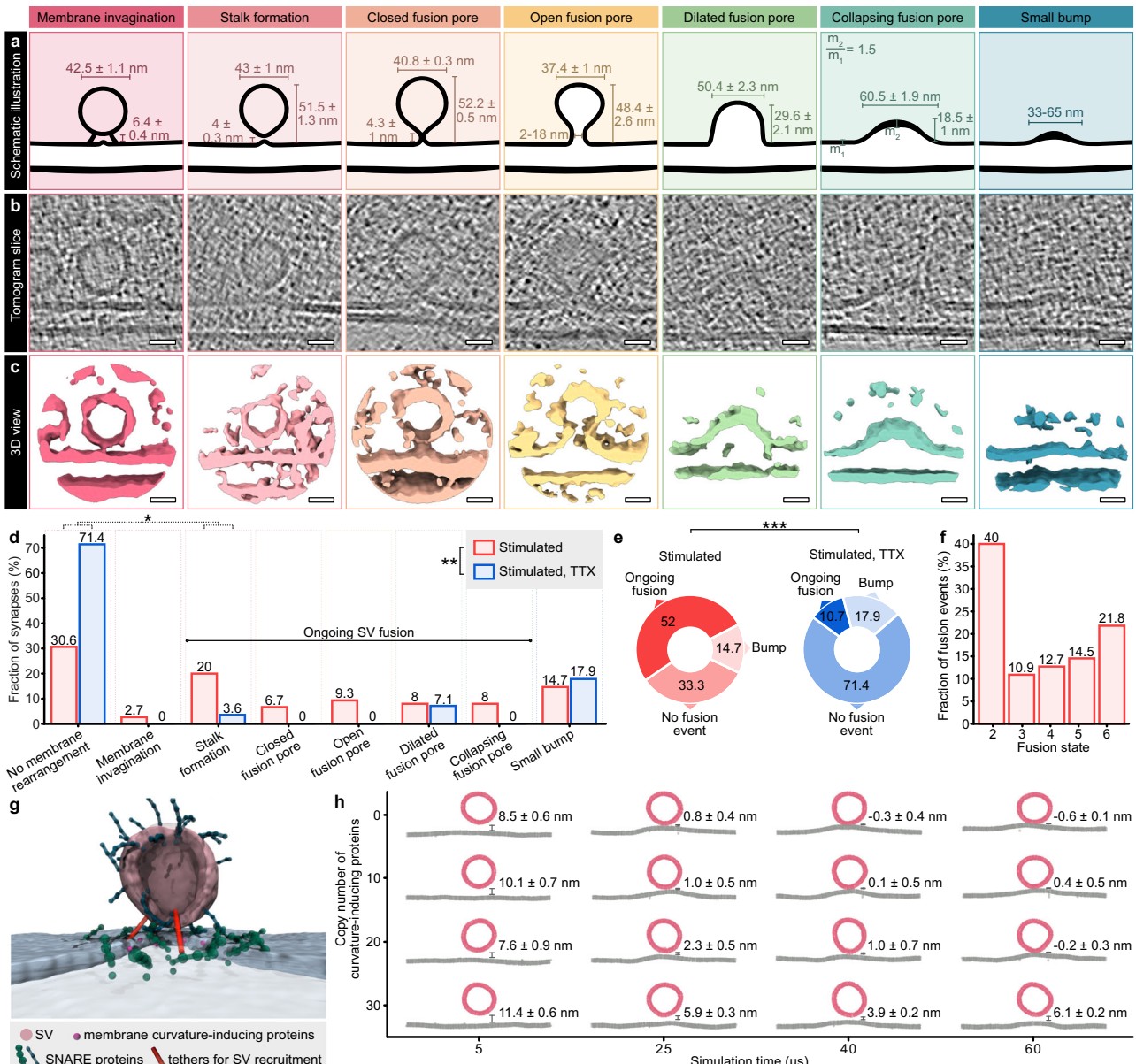

**Fig. 3 | Cryo-ET of synaptic vesicle fusion states. a–c** Schematic illustrations with details about size measurements (**a**), exemplary cryo-ET slices (**b**), and isosurfaces (**c**) of 7 categories of membrane rearrangements observed at synaptic active zones of stimulated neurons. Scare bars: 20 nm. **d** Fractions of synapses with or without membrane rearrangements in stimulated and stimulated, TTX-treated samples. Fisher's exact test (stimulated vs. stimulated, TTX for all 8 categories) $p = 0.007$; Fisher's exact test with Benjamini-Hochberg correction for multiple comparisons (no membrane rearrangements vs. stalk formation, stimulated vs. stimulated, TTX) $p = 0.035$. **e** Fractions of synapses without membrane rearrangements, ongoing fusion, or bumps in stimulated and stimulated, TTX-treated samples. Synapses were counted as "ongoing fusion" if at least one stalk formation, closed, open,

dilated, or collapsing fusion pore was observed. Fisher's exact test $p < 0.001$. **d**, **e** Stimulated sample: $N = 75$ synapses from 3 grids and 2 independent freezings, TTX sample: $N = 28$ synapses from 1 grid. **f** Fractions of individual fusion states in stimulated synapses. $N = 55$ fusion events. **g** Snapshot of a coarse-grained simulation of an SV approaching the active zone membrane, with particle-based representation of recruiting tethers, SNARE proteins and varying copy numbers of proteins inducing membrane curvature at the active zone below the SV. **h** Time evolution of the vertical distance between SV and active zone membrane, depending on the copy number of membrane curvature-inducing proteins. Source data are provided as a Source Data file.

approximation (in our model, SV fusion was not enabled). This means that although a slight invagination of the AZ membrane, as observed by us and in a previous study[22], may precede SV fusion, it is unlikely induced by proteins like synaptotagmin-1 but rather a consequence of SNARE zippering[7,9].

**Depletion of membrane-proximal SVs in stimulated synapses**
In addition to ongoing SV fusion events, we analyzed the distribution of tethered SVs within a distance of 24 nm from the AZ membrane (Fig. 4a, b). In previous EM studies, changes in the abundance of

membrane-near SVs have been used as confirmation for successful action potential induction and subsequent release[10–13,37,49]. To test whether we could reproduce these findings with our workflow, we first compared all stimulated synapses ($n = 515$ SVs from 54 synapses) to TTX-treated synapses ($n = 162$ SVs, 19 synapses), whereby synapses with only bumps (state 7) were not included. While the total number of SVs within a distance of 24 nm was not significantly different between the two groups (Mann-Whitney test $p = 0.477$, $U = 456$, Fig. 4c), we counted on average $1.4 \pm 0.6$ fewer SVs per AZ within a maximum distance of 6 nm (Mann-Whitney test $p = 0.035$, $U = 352.5$, proximal

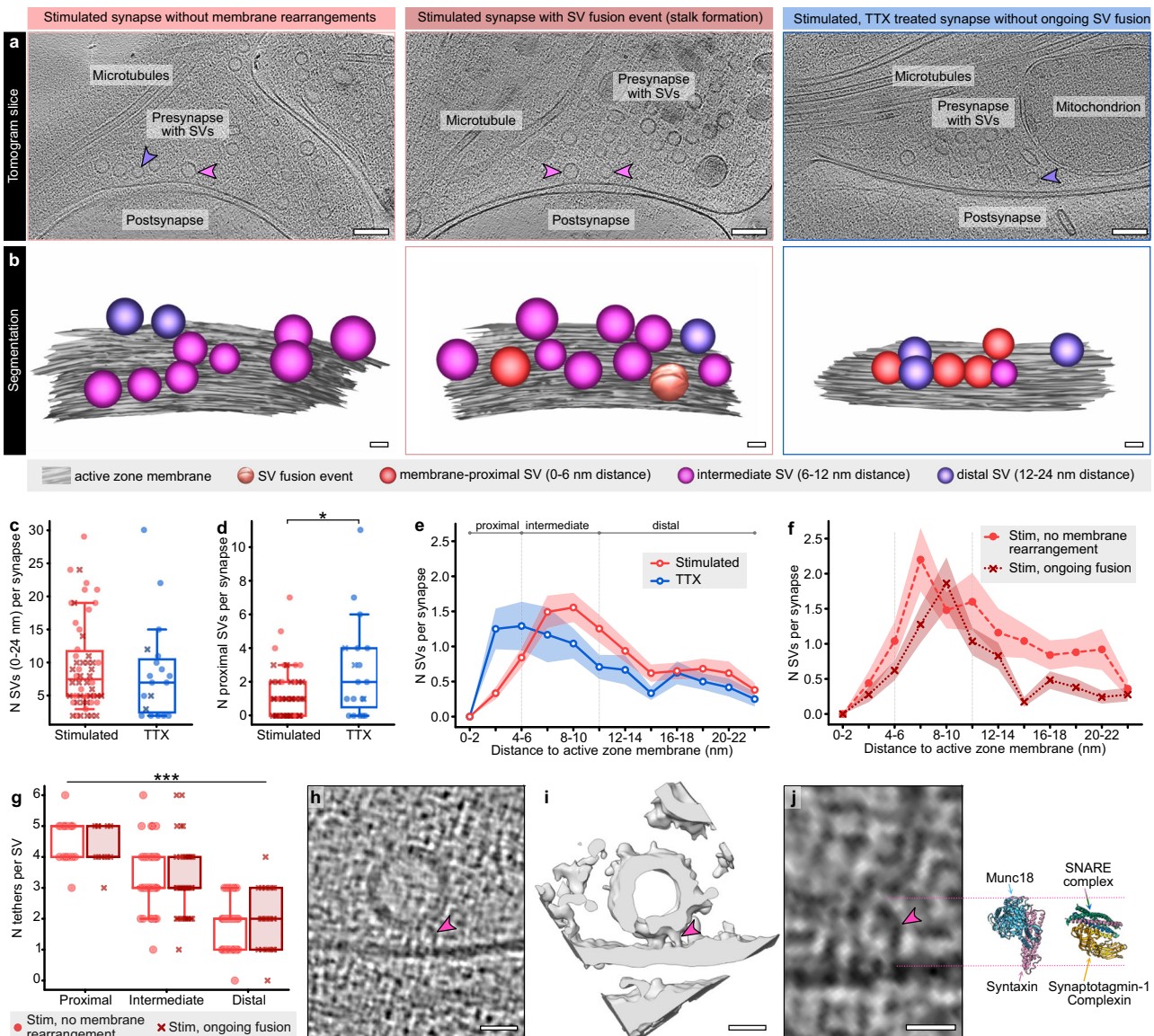

**Fig. 4 | Stimulation-induced changes in the distribution and tethering of membrane-near SVs. a** Exemplary cryo-ET slices of synapses without and with ongoing fusion in stimulated and stimulated, TTX-treated neurons. Scale bars 100 nm. **b** Manual segmentations of active zones and membrane-near SVs. Scale bars 20 nm. **c**, **d** Numbers of SVs per synapse with a max. distance of 24 nm (**c**) or 6 nm (proximal SVs, **d**). stim: $N = 54$ synapses, TTX: $N = 19$ synapses. **c** Two-tailed Mann-Whitney test, $p = 0.477$, $U = 456$. **d** Two-tailed Mann-Whitney test, $p = 0.035$, $U = 352.5$. **e** Distribution of membrane-near SVs in stimulated and stimulated, TTX-treated neurons. **f** Distribution of membrane-near SVs in synapses of stimulated neurons with ongoing fusion and without membrane rearrangements. No membrane rearrangement: $N = 25$ synapses, ongoing fusion: $N = 29$ synapses. **g** Numbers of tethers for membrane-proximal, intermediate (6–12 nm) and distal (1–24 nm)

SVs of stimulated neurons. No membrane rearrangement: $N = 145$ SVs from 15 synapses, ongoing fusion: $N = 114$ SVs from 18 synapses. Kruskal-Wallis test, $p < 0.001$, KWS = 140. **h** Exemplary tomogram slice of a multi-tethered SV. The pink arrowhead indicates one of the tethers. Scale bar 20 nm. **i** Isosurface of the same tethered SV. Scale bar 20 nm. **j** Atomic models of syntaxin/Munc18 (left, PDB: 4JEU) and an assembled SNARE complex with synaptotagmin-1 and complexin (right, PDB: 5W5D) in comparison to the tether indicated in (**h**) for size estimation, scale bar 10 nm. Light blue: Munc18, pink: syntaxin-1, green: SNAP-25, dark blue: synaptobrevin-2, yellow: complexin, orange: synaptotagmin-1. Box plots indicate median, 25% and 75% quartiles, whiskers indicate 10–90% percentiles. Data in line graphs are represented as mean ± SEM. Source data are provided as a Source Data file.

SVs, Fig. 4d, e) in stimulated synapses. Between 6 and 12 nm (intermediate SVs), we observed slightly more SVs in stimulated synapses (Supplementary Fig. 11c). A comparable redistribution of SVs was reported using "zap-and-freeze" and freeze substitution, however, the effects were more drastic (loss of approx. 40% of all "docked" SVs due to SV fusion and putative undocking)[10].

In addition, we analyzed SV distributions of stimulated synapses with ongoing SV fusion ($n = 29$) and without membrane rearrangements ($n = 25$) individually. In the membrane-proximal SV pool (0–6 nm distance), we observed $0.6 \pm 0.4$ fewer SVs per AZ in

stimulated synapses containing at least one fusion event (Fig. 4f). Consequently, also the subgroup of stimulated synapses without observed ongoing SV fusion had on average less membrane-proximal SVs than TTX-treated synapses (counted difference $1.1 \pm 0.7$ SVs, Supplementary Fig. 11b, e). It is likely that this subgroup of stimulated synapses without membrane rearrangements consists of synapses postfusion (neurotransmitter release has already taken place) and synapses without neurotransmitter release (non-releasing synapses). Assuming that the distribution of membrane-proximal SVs in postfusion synapses is comparable to synapses with ongoing fusion, as

reported recently for synapses up to 11 ms after action potential induction[10], whereas non-releasing synapses assumingly resemble TTX-treated synapses[10,50], we calculated the theoretical fraction of non-releasing synapses (see Supplementary Methods): Within the group of synapses without membrane rearrangements, the fraction of non-releasing synapses would be 17%. Of all stimulated synapses, the fraction of non-releasing synapses would be 8%, resulting in a theoretical synaptic release probability of 92% with our workflow. The release probability of excitatory hippocampal synapses at a calcium concentration of 4 mM and near-physiological temperature was reported to be ~ 85%[51], as also confirmed by our iGluSnFR3 response rate for electrically stimulated neurons of 86% (Source Data). Taking into account that action potentials could be induced in 80-90% of optogenetically stimulated neurons (Supplementary Fig. 1c), the expected synaptic release probability for our setup is 68–77% and thus even lower. In other words, we observed on average fewer membrane-proximal SVs in the subgroup of stimulated synapses without ongoing fusion than expected. Kusick and colleagues[10] came to the same conclusion and attributed the low SV number to transient SV undocking during or shortly after fusion.

Previous work has shown that not only the distribution of SVs, but also the number of tethers dynamically changes during synaptic activity, whereby the formation of three or more tethers connecting SV and AZ membrane was suggested to be a morphological correlate of priming and a prerequisite for SV fusion[21,22]. Although our tomograms of synapses were comparatively thick, going along with a potentially worse signal-to-noise ratio than FIB-milled samples or purified synaptosomes, we were able to quantify tethers in our stimulated samples. Overall, we manually quantified tethers at 114 SVs from 18 synapses with ongoing fusion and 145 SVs from 15 synapses without membrane rearrangements. In both groups, we observed a linear correlation of tether number and distance between SV and AZ membrane (Supplementary Fig. 11h), whereby membrane-proximal SVs had the highest average number of tethers (4.6 ± 0.1 and 4.4 ± 0.2 for synapses without and with fusion event, respectively) and distal SVs the lowest (Fig. 4g). The number of tethers below SVs with slight membrane invagination was 4.2 ± 0.3 and did thus not differ from the other SVs within the same distance to the cell membrane. Of note, we observed short, vertical tethers predominantly below SVs (Fig. 4h and Supplementary Fig. 12) and longer, angular and curved tethers predominantly at the sides of membrane-proximal SVs (Supplementary Fig. 12). For size estimation, we positioned atomic models of different exocytic proteins/complexes next to tethered SVs (Fig. 4j and Supplementary Fig. 12). Based on their size, SNARE proteins may be involved in the formation of the short tethers observed here, as suggested previously[15,22,23]. However, not only assembled SNARE complexes together with synaptotagmin-1 and complexin (PDB: 5W5D[6]), which would indicate a primed state of the SV, are likely candidates. Syntaxin-1 and Munc18 (PDB: 4JEU[52]) or syntaxin-1, synaptobrevin-2 and Munc18 (PDB: 7UDB[53]), both representing states preceding SV priming, would fit equally well. Size-wise, Munc13 (PDB: 7T7V or 7T7X[54]) could be involved in the formation of longer angled tethers, however, we cannot rule out that Munc13 is also part of short vertical tethers. Of note, we also observed filaments connected to SV fusion intermediates, e.g., around the space between SV and AZ membrane during stalk formation (Supplementary Fig. 8c). Whether these filaments resemble parts of the SV fusion machinery, e.g., assembled SNARE complexes, needs to be investigated further.

### Multivesicular release and release site refilling

In 23% of all stimulated synapses with ongoing SV fusion (9/39 synapses), we observed multiple fusion events (Fig. 5a, c), whereby most of these synapses contained two (Fig. 5d). Likewise, we observed multiple small bumps per synapse in 55% of synapses containing bumps (6/11 synapses, Fig. 5b, e). At the synapses with multivesicular release

(MVR), the individual fusion events did not preferentially fall into the same category.

Particularly during fast and sustained neurotransmitter release, release sites need to be refilled with SVs. In our synapses, we observed that SVs were not only connected to each other via pleomorphic interconnectors, but also partially connected to fusion intermediates of different categories (Fig. 5f, also see refs. 21,22). Considering that filamentous connections between membrane-near SVs were shown to persist or even increase during SV fusion[21,22], it is likely that also the linkers between SVs and fusion events are stable and/or strengthened during action potential-induced calcium influx. This way, fusing SVs may directly recruit new SVs for very fast release site refilling. We indeed observed filamentous interconnectors at 62% of fusion events (26/42). To test whether these interconnectors are involved in the recruitment of nearby SVs during fusion, we performed a nearest neighbor analysis. We first measured distances between fusion events and nearest neighbor SVs, as well as distances between membrane-proximal SVs (max. distance of 6 nm to the AZ membrane) and their respective nearest neighbor SVs in the groups of stimulated synapses without membrane rearrangements and stimulated synapses with ongoing fusion (Fig. 5g–i). We compared average distances for membrane-proximal SVs and fusion events per synapse (Fig. 5g), resulting in a nonsignificant trend towards greater nearest neighbor distances in synapses with ongoing fusion. The direct synapse-wise comparison of distances between SVs and between fusion events and SVs (Fig. 5h) revealed no difference between the two groups. We further compared nearest neighbor distances in membrane-proximal SVs and fusion events individually, revealing significantly lower nearest neighbor distances for membrane-proximal SVs in synapses without membrane rearrangements than in synapses with ongoing fusion (Kruskal-Wallis test, $p < 0.001$, KWS = 14.55). We attribute this difference primarily to the overall lower number of membrane-near SVs in the group of stimulated synapses with ongoing fusion. Interestingly, fusion events without and with interconnectors formed two subpopulations, whereby fusion events without interconnectors were associated with significantly larger nearest neighbor distances (Mann-Whitney test, $p < 0.001$, U = 26.5). Moreover, we noticed that fusion events with interconnectors were more prevalent in earlier fusion states, whereas fusion events without interconnectors were primarily associated with collapsing fusion pores (Fig. 5j, two-way ANOVA, $p < 0.001$, $F = 20.14$ for interconnected vs. not interconnected fusion events). Together, these findings point towards a role of filamentous interconnectors in direct and immediate release site refilling, particularly in the early phase of SV fusion.

## Discussion

In summary, we have developed a workflow combining optogenetic stimulation of neurons with plunge freezing and cryo-ET to achieve both high temporal and structural resolution while preserving the near-native cellular architecture during near-physiological stimulation. Although optogenetic plunge freezers have been introduced before[55–57], we now demonstrate their suitability for in situ applications. In comparison to other approaches for time-resolved cryo-EM, typically involving the mixing and spraying of reactants onto EM grids[38,58], we achieved a similar temporal resolution with optogenetics: The minimal delay between the end of a light pulse and freezing in our system was only 2-3 ms, while microfluidic spraying techniques achieve up to 10 ms and on-grid mixing approaches achieve less than 5 ms[58]. Considering that the optical fibers can be positioned along the path of the tweezers and that the duration of the light pulse(s) can be modified, our system provides high flexibility and high precision, possibly in the sub-millisecond range. The coupling of light pulse and cryofixation may thus be of interest for both in situ experiments and in vitro applications.

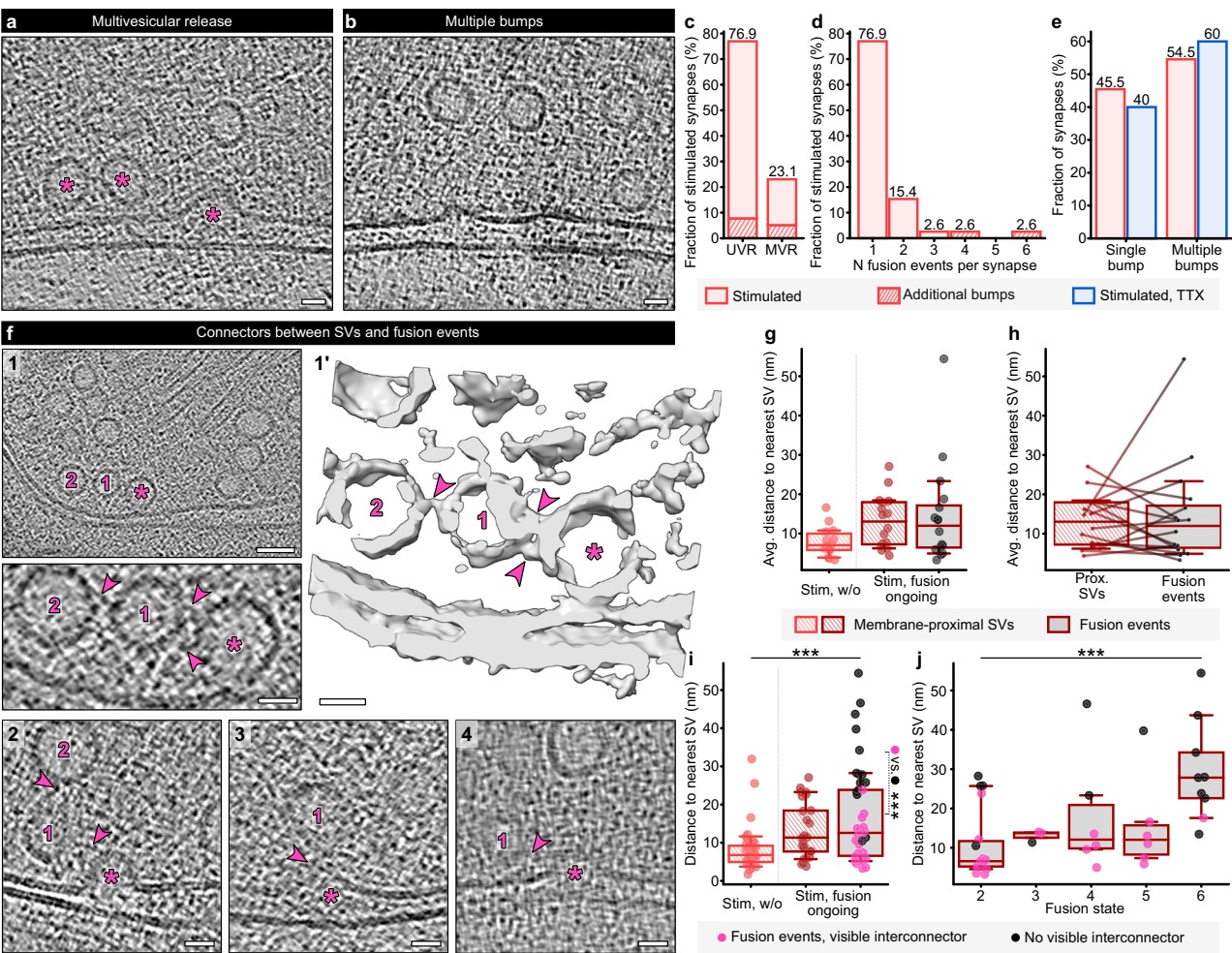

**Fig. 5 | Multivesicular release and filamentous interconnectors mediating SV resupply. a** Exemplary tomogram slice of multivesicular release (MVR). The asterisks indicate two stalks and a dilated fusion pore at one active zone. Scale bar 20 nm. **b** Exemplary tomogram slice of multiple bumps. Scale bar 20 nm. **c** Fractions of synapses with univesicular release (UVR) and MVR in stimulated synapses. Synapses additionally containing bumps are indicated as shaded areas. **d** Numbers of fusion events per synapse in the stimulated sample. **e** Fractions of single and multiple bumps in stimulated and stimulated, TTX-treated neurons. Stimulated sample: $N = 75$ synapses, TTX sample: $N = 28$ synapses. **f** Examples for SVs connected to fusion events. 1: tomogram slice and zoom-in (lower panel), 1′: isosurface of a fusion stalk with two additional SVs. Asterisks label fusion events, numbers display connected SVs, arrowheads display connecting filaments. Scale bars 20 nm. 2–4: Exemplary tomogram slices of dilating (left and middle panels) and collapsing (right panel) fusion pores connected to SVs. Scale bars 20 nm. **g** Distances of membrane-proximal SVs and fusion events to nearest neighboring SVs, averaged per synapse. Stimulated without membrane rearrangements: $N = 14$ synapse, stimulated with ongoing fusion: $N = 16$ synapses. Kruskal-Wallis test,

$p = 0.086$, KWS = 4.908. **h** Synapse-wise comparison of average distances between membrane-proximal SVs and neighboring SVs or fusion events and neighboring SVs in the group of stimulated synapses with ongoing fusion, $N = 16$ synapses. Wilcoxon test of pairs, $p = 0.86$. **i** Individual distances of membrane-proximal SVs and fusion events to nearest neighboring SVs. Stimulated synapses without membrane rearrangements: $N = 37$ membrane-proximal SVs, stimulated synapses with ongoing fusion: $N = 25$ membrane-proximal SVs and $N = 42$ fusion events. Kruskal-Wallis test, $p < 0.001$, KWS = 14.55. Additional comparison of distances for fusion events with (pink, $N = 26$) and without (black, $N = 16$) visible interconnectors. Mann-Whitney test, $p < 0.001$, U = 26.5. **j** Distances between fusion events and neighboring SVs for fusion states 2–6. Fusion events with visible interconnector: $N = 26$, fusion events without visible interconnector: $N = 16$. Two-way ANOVA, comparison of fusion states $p = 0.08$, F = 2.28, comparison of interconnected vs. not interconnected $p < 0.001$, F = 20.14. Box plots indicate median, 25% and 75% quartiles, whiskers indicate 10–90% percentiles. Source data are provided as a Source Data file.

When compared to existing workflows designed to visualize SV fusion[10–13,21,22,37,49], our timed in situ cryo-ET workflow shows distinct advantages. Several studies combined optogenetic or electrical stimulation with high-pressure freezing. While in particular electrical stimulation is considered more temporally precise than optogenetics because of the direct depolarization of synaptic boutons, the overall delay between stimulation and cryofixation is higher due to greater sample thickness (3-4 ms[10] vs. < 1 ms for plunge freezing[38]). In addition, high-pressure frozen samples in these studies underwent freeze substitution and resin embedding. While this enabled large-scale morphometric analyses across hundreds of synapses, the biologically interpretable resolution is substantially reduced by heavy metal

staining and sectioning. This makes it nearly impossible to distinguish stalk formation or closed fusion pores from membrane-near SVs. Consequently, these studies primarily reported on "fusion pits"[10], resembling wide open or dilating fusion pores in our study.

Only a few cryo-ET studies have addressed SV fusion directly. In two recent studies using KCl for chemical stimulation of synaptosomes, examples of SV fusion intermediates (categories 2-7 according to our definition) were qualitatively assessed[21,22] and categorized as early and late fusion events[22]. In addition, spontaneous SV fusion events were described in 7% of synapses in a recent cryo-ET study investigating mouse brain tissue[19]. With our workflow, we now show action potential-induced SV fusion events in intact neurons. While the

SNARE-mediated SV fusion process itself is supposedly comparable in synaptosomes and intact neurons, we were able to reliably induce individual action potentials reflecting physiological neuronal behavior and directly confirm successful stimulation using a fluorescent glutamate sensor. Beyond that, studying intact neurons allowed us to characterize synapses within their native cellular organization, including not only an intact cytoskeleton but also the positioning of organelles like mitochondria and ER. The large number of synapses and fusion events we captured enabled a quantitative analysis of morphometrically distinct fusion intermediates, which in turn formed the basis for our coarse-grained simulation of SV fusion initiation and will possibly provide morphological parameters for future in silico studies of SV fusion.

The mechanism by which SV fusion initiates has long remained unclear, largely due to its rapid dynamics and fine ultrastructural changes, which are inherently hard to capture with conventional imaging techniques. While there is a general consensus about the opening and collapsing of fusion pores, the mechanisms behind fusion initiation are still under debate and may vary between different cellular processes[3]. Our observations favor a model in which SV fusion in neuronal synapses is initiated by stalk formation, whereby the AZ membrane is only slightly invaginated when the spherical SV converts into a droplet shape. At this stage, the SV and cell membrane remain ~4 nm apart, while a tip-like connection is formed and lipids of the outer membrane leaflets start to intermix. Similar observations were made by Radecke et al.[22], who likewise identified three examples of early fusion events resembling our category of stalk formation. Given that we only observed one example of a tightly docked SV at an active zone, we do not consider the formation of a hemifusion diaphragm as an integral step during action potential-induced fusion. Instead, the tightly docked SV may display a morphological correlate of a spontaneous, likely calcium-independent, SV fusion event[26]. Alternatively, the SV may have been trapped at the membrane without being fusion competent.

Compared to later fusion states, stalk formation appeared more frequent, likely due to the comparatively high energy barrier before fusion pore formation, when the inner leaflets of the SV and cell membrane need to fuse[45]. In addition, stalk formation may be reversible, meaning that not all observed stalks would have led to full SV fusion[44,45]. Although we and others[22] observed slight AZ membrane invaginations below tethered SVs, we do not have experimental evidence that they directly lead to SV fusion. MD simulations described membrane invaginations preceding SV fusion and attributed them to membrane curvature-inducing functions of proteins such as synaptotagmin-1[59,60]. However, a direct role of synaptotagmin-1 in inducing membrane curvature has been questioned recently[7,9]. Our coarse-grained simulation likewise indicated that the induction of membrane curvature does not appear to support fusion initiation. Instead, the observed slight membrane invaginations may result from SNARE complex zippering, as recently shown in an MD simulation of SNARE-mediated SV fusion without synaptotagmins[61].

Beyond mediating membrane fusion, exocytic proteins were shown to be involved in the recruitment and priming of SVs[39,62]. Overall, our findings support the idea that multiple tethers are preferentially formed between the AZ membrane and SVs in close membrane proximity[21–23]. Yet, we neither found differences in tether numbers between synapses with and without ongoing SV fusion nor between SVs with and without membrane invaginations. Considering that we observed fewer SVs in stimulated synapses primarily within a distance of 6 nm from the AZ membrane and membrane invaginations below SVs with an average distance of 6.4 nm, it is likely that most functionally primed SVs are also located here. Consequently, not the number of tethers but rather their molecular composition may correlate with the priming state of SVs[2].

We further observed fewer membrane-proximal SVs in stimulated synapses without ongoing fusion than expected. This could be explained by intrinsic differences in the organization of SV pools between releasing and non-releasing synapses. However, previous studies showed that numbers of docked SVs were comparable between synapses of different hippocampal neurons, which have drastically different release probabilities[50]. Taking into account that our observed probability for MVR and our calculated release probability were lower than described for comparable stimulation conditions[51], a high fraction of SVs may have fused already before cryofixation. This suggests that synapses with very fast responses (too fast to be captured with our setup) also preferentially showed higher fractions of MVR, leading to a stronger depletion of the membrane-proximal SV pool. Also, a transient dissociation of proximal SVs from the AZ membrane is conceivable, as previously suggested[10].

Interestingly, we observed SVs linked to fusion events via filamentous interconnectors in a fraction of our tomograms, whereas in others, the space around fusion events was not occupied by SVs. Assuming that filamentous connections to SVs may strengthen[22] or destabilize during synaptic activity, membrane-proximal SVs could either be recruited by fusing SVs, AZ scaffolds and cytoskeletal filaments for release site refilling, or pulled away from the membrane towards the cytosolic SV pool. This dynamic redistribution of membrane-proximal SVs, likely mediated by filamentous connectors, may represent a mechanism for direct, activity-dependent SV replenishment. Considering that we primarily observed differences in nearest neighbor distances between non-releasing vs. releasing synapses, rather than between SVs and fusion events, and that the majority of fusion events contained interconnectors, it is indeed likely that these filaments persist during fusion initiation and thus contribute to immediate SV replenishment. The lower nearest neighbor distance in synapses without ongoing fusion is likely a consequence of the observed higher number of membrane-near SVs in this group. However, taking into account that the majority of late fusion events did not contain an interconnector, it may be speculated that some of the connections disassemble during fusion. Such a mechanism could protect the AZ from overcrowding and provide an elegant explanation for the discrepancy in SV numbers observed in Fig. 4f and Supplementary Fig. 11, as well as for the previously observed transient undocking of membrane-near SVs during fusion[10].

Together, our study does not only display one of the first cell biological applications of a light-coupled plunge freezer but also provides an in-depth characterization of SV fusion in situ. Nevertheless, we wish to address the following limitations of our study: The fractions of fusion events categorized as state 5, dilating fusion pore, were similar in stimulated and TTX-treated groups. We assume that this is a stochastic effect due to the low total number of synapses in the TTX group. Taking into account that SV fusion intermediates have been observed in 7% of unstimulated synapses in a recent cryo-ET study[19], the two dilating fusion pores observed in TTX-treated synapses may stem from spontaneous fusion events. However, we cannot rule out that also endocytic events occasionally take place directly at the AZ and match our definition of fusion events (approx. size of an SV, no visible clathrin coat, directly at the AZ), leading to an overcounting of observed fusion events. Apart from the classification of fusion events, also the quantification of filamentous tethers and interconnectors in our study is empirical. Although we minimized bias, e.g., by parallel analysis of groups, the manual quantification of filaments is subjective. A comparison of absolute tether numbers from this and previous studies[20–23,63] is therefore not meaningful. Further improvements in cryo-ET data acquisition and (post-) processing, increasing the signal-to-noise ratio of in situ data from synapses, are necessary for reliable automated tether recognition and potentially also identification of proteins involved in tether formation. Lastly, we utilized numbers of fusion events across individual fusion states as a relative

readout for inferring the kinetics of fusion progression and for the generation of Markov state models. Although our models are well in line with previous studies reporting e.g., on the metastable and reversible nature of stalk formation[44,45], our results need to be interpreted with caution: While we analyzed at a single time point after stimulation, we observed at this time point a wide range of distinct action potential dependent structures, arguing for extensive structural dynamics within the few milliseconds following fusion initiation. However, we must predominantly speculate about the order of fusion intermediates. To better dissect the time course of SV fusion, multiple time points after action potential induction would be required. For our Markov state models, we further assumed that all transitions following stalk formation are irreversible and that fusion progresses according to our classification; however, these assumptions lack direct experimental validation. Finally, we implicitly assume that the fusion progression and kinetics are the same for each individual SV. This is likely not the case, as factors like the number of formed SNARE complexes may influence fusion initiation and progression.

## Methods

### Mass culture of mouse primary hippocampal neurons

**Astrocyte feeder culture.** Astrocytic and neuronal mass cultures were prepared from P0-P2 C57/BL6/N mice of either sex. To prepare astrocyte feeder layers, mice were decapitated, and cortices were isolated in cold HBSS-HEPES. The tissue was digested in 0,05% trypsin-EDTA for 15–20 min at 37 °C followed by manual trituration. The isolated astrocytes were transferred to T75 flasks and cultured for two weeks in Dulbecco's modified Eagle medium supplemented with 10% fetal calf serum (FCS), 100 U/ml penicillin and 100 µg/ml streptomycin (DMEM) at 37 °C and 5% $CO_2$. Confluent astrocytes were trypsinated (0,05% trypsin-EDTA) and seeded on collagen/poly-D-lysine coated coverslips (for RT experiments) or coated wells (for cryo experiments and banker cultures) in a density of $7.5 \times 10^4$ cells/well (six-well plates). The astrocytes were cultured for an additional week before FUDR (8.1 mM 5-fluoro-2-deoxyuridine and 20.4 mM uridine in DMEM) was added to arrest glia proliferation.

**Neuronal culture on coverslips.** For live imaging, RT confocal imaging and electrophysiological recordings, primary hippocampal neurons were used as co-cultures with astrocytes or separated from astrocyte feeder layers as banker cultures. Since we did not find differences in the responsiveness of neurons in the two different culture systems, we pooled data from mass cultures and banker cultures. To isolate neurons, hippocampi were dissected in cold HBSS-HEPES and digested using 20 U/ml papain for 45 min at 37 °C, followed by manual trituration. For co-cultures, isolated neurons were seeded directly on 1-2 weeks old astrocyte feeder layers in a density of $3 \times 10^4 - 5 \times 10^4$ neurons/well. For banker cultures, neurons were seeded on coverslips coated with collagen/poly-D-lysine and ornithine ($5 \times 10^4$ neurons/well), and the coverslips were transferred to well plates containing astrocyte feeder layers after neurons were allowed to adhere to the coverslips for 1 h. Neurons were cultured for 14–18 days at 37 °C in neurobasal-A medium containing 2% B-27, 1% Glutamax, 100 U/ml penicillin and 100 µg/ml streptomycin (NBA), lentiviruses and AAVs were added on DIV2-3.

**Neuronal culture on EM grids.** For culturing neurons on EM grids, mesh pedestals were 3D printed (see Source File 1 for a printing scheme for 6-well and 12-well plates) and placed on top of astrocyte feeder layers. DMEM medium was replaced by NBA medium. Quantifoil R3.5/1 AU holey carbon grids (400 mesh, 200 mesh and 200 mesh finder grids) were cleaned with chloroform and acetone, followed by glow-discharging. Directly afterwards, the grids were placed on droplets of collagen/poly-D-lysine coating solution and incubated for 20 min under UV light and 1-2 h at 37 °C. The grids were washed in PBS overnight and then transferred to the well plates containing the astrocyte feeder layers and pedestals, 3 grids per well. Primary hippocampal neurons were seeded in a density of $1.5 \times 10^5$ - $2 \times 10^5$ cells/well, viruses were applied on DIV2-3.

### Live imaging of biosensors

Live imaging of neurons expressing the glutamate sensors iGluSnFR, iGluSnFR3-PDGFR, or iGluSnFR3-GPI was performed at elevated temperature (~32-34 °C) using an inverted microscope (Olympus IX51) with a custom-built in-line heating system and a 60 x water immersion objective with a heated collar (Warner Instruments). Cells were perfused with high-calcium extracellular solution containing the following: 140 mM NaCl, 2.4 mM KCl, 10 mM HEPES, 10 mM glucose, 4 mM $CaCl_2$, and 1 mM $MgCl_2$ (300 mOsm; pH 7.4). 6 µM NBQX, 30 µM bicuculline, and 10 µM AP-5 were added to block neuronal network activity of the mass culture. Action potentials (2 ms depolarization) were induced using a field stimulation chamber (Warner Instruments), Multiclamp 700B amplifier, and an Axon Digidata 1550B digitizer controlled by Clampex 10 software (all Molecular Devices). To assess the suitability of the biosensors for our cryo-ET workflow, two action potentials with an inter-stimulus interval of 25 ms were induced, which corresponds to the minimal time between the first and second stimulus of the optogenetic plunge freezer (limitation due to the bottom of the incubation chamber of the Vitrobot). Samples were illuminated by a 490 nm LED system (CoolLED) with an exposure time of 10 ms, images were captured with an Andor iXon Life 897 camera using Andor Solis v4.32 (Oxford instruments) at a frame rate of 40 fps. The imaging was performed blind.

### Optogenetic stimulation and vitrification

For optogenetic stimulation of neurons, a plunge freezer (Vitrobot Mark IV, Thermo Fisher Scientific) was equipped with a high-intensity LED (Schott LLS3, wavelength 470 nm), from which one PMMA optical fiber with PVC insulation (inner diameter 3 mm) reached inside and one below the chamber of the Vitrobot. The end points of the optical fibers were installed with a distance of max. 5 mm to the path of the plunge frozen EM grids and illuminated the grids entirely. The LED was controlled by an infrared sensor that detected the downward motion of the tweezer holding the EM grid. The grid was illuminated twice for 5 ms: the first light pulse was applied within the chamber, max. 100 ms before freezing, and the second light pulse was started approximately 7 ms before the sample reached the liquid ethane. The correct timing and illumination were confirmed using the super-slow-motion mode of a Samsung Galaxy S20 camera with a frame rate of 960 fps.

At DIV16-18, each EM grid containing neurons was briefly washed in a high-calcium solution containing 140 mM NaCl, 2.4 mM KCl, 10 mM HEPES, 10 mM glucose, 4 mM $CaCl_2$, 1 mM $MgCl_2$, 3 µM NBQX, and 30 µM bicuculline (~300 mOsm; pH7.4) pre-warmed to 37 °C and directly transferred to the plunge freezer. 4 µl of high-calcium solution supplemented with 10 nm BSA-gold (Aurion, OD ~2) were applied on the grid prior to blotting for 12–16 s (backside blotting, blot force 10) at 37 °C and a relative humidity of 80%. The grids were plunge-frozen in liquid ethane and stored in liquid nitrogen until further use. For TTX-treatment, grids were incubated in high-calcium solution containing 1 µM TTX for 1-2 min prior to freezing.

### Cryo-confocal microscopy

**Data acquisition.** After plunge freezing, EM grids were clipped into autogrids and transferred to a TCS SP8 cryo-confocal microscope equipped with a 50x CLEM cryo-objective, NA 0.9 (Leica Microsystems). Overviews of each grid were acquired in brightfield and fluorescence mode; the reflective mode was used on a subset of grids to estimate the ice thickness. The iGluSnFR fluorescence signal was used to confirm overall successful grid stimulation. For the comparison of iGluSnFR fluorescence intensities without and with stimulation,

only the fluorescence signal of the fluorophore attached to ChR2 (YFP or mScarlet) was used to select regions of interest for subsequent cryo-confocal microscopy to avoid bias. Cryo-confocal stacks (z-steps 0.5 μm, pixel size 0.11 μm) were acquired with optimized filter settings of the HyD detector to minimize YFP or mScarlet crosstalk with the GFP signal. From grids intended for correlative confocal and electron microscopy, only a few confocal stacks were acquired, and the overall acquisition time per grid was limited to 30 min to avoid strong ice contaminations.

**Analysis of cryo-confocal microscopy and correlation with cryo-electron tomography.** Fluorescence intensities in cryo-confocal stacks of unstimulated, stimulated, and TTX-treated plunge-frozen neurons were compared using fiji software v1.54[64]. Maximum intensity z-projections of each confocal stack were generated, and 300 × 300 pixel regions of interest (ROIs) containing individual neurites were extracted. The mean fluorescence intensity (Fig. 2d) was measured per ROI and averaged per confocal stack. To compare the distribution of fluorescence intensities (Fig. 2c), fluorescence intensity histograms were generated for each ROI. A threshold of 15 was applied for background subtraction, and the pixel counts per intensity were normalized to the total pixel number per ROI. These relative intensity histograms of ROIs were averaged per confocal stack. The threshold of 70 for high-intensity pixels was visually assessed in stimulated samples. The fraction of pixels > 70 (Fig. 2e) was calculated per ROI and averaged per confocal stack. The correlation of fluorescence and TEM (Fig. 2f) was performed manually using the navigator of the Leica lasx software and fiji.

**Cryo-electron tomography**

*Data acquisition.* Cryo-ET data collection of optogenetically stimulated neurons cultured on EM grids was performed on a Titan Krios G3i electron microscope (Thermo Fisher Scientific) equipped with a K3 direct electron detector with BioQuantum energy filter (Gatan) and operated at 300 kV. Tilt series were typically acquired with 10 frames per tilt at a magnification of 15,000 x and a pixel size of 3.2 Å in superresolution mode using SerialEM[65] and PACE-tomo[66]. Tilt angles ranged from −50° to +50° and 2° angular increment in a dose-symmetric[67] tilt-scheme. The defocus values ranged from -3 to -6 μm and the total electron dose was 106–125 e⁻/Å².

**Tomogram reconstruction.** Tomograms used for analysis were reconstructed semi-automatically using the tomoBEAR[68] pipeline: Aligned frames were motion-corrected using MotionCor2[69]. The tilt series alignment was performed by DynamoTSA[43] and manually refined using 10 nm gold fiducial markers in IMOD[70,71]. For each projection, defocus values were measured by Gctf[72], and CTF correction was performed using the IMOD command ctfphaseflip[73]. Four-times binned 3D reconstructions (final pixel size 12.28 Å) from CTF-corrected, aligned stacks were obtained by weighted back projection in IMOD.

In total, 312 tilt series of stimulated neuronal samples and 95 tilt series of TTX-treated neuronal samples were reconstructed with tomoBEAR and manually screened for synapses, which were only recognizable after reconstruction. We visually identified synapses as presynaptic boutons filled with SVs and a synaptic AZ, a synaptic cleft of 10–30 nm width, and a postsynaptic bouton with visible postsynaptic density. Based on these criteria, we identified 75 synapses in the stimulated samples and 28 synapses in the TTX-treated sample that were used for further analysis. For cryo-ET analyses of stimulated synapses, we did not analyze each freezing/grid individually but pooled synapses of three grids/two freezings because we did not note any significant differences in numbers of SVs or putative fusion events between them.

**Segmentation and analysis of cryo-electron tomography data**
**Contrast enhancement.** The contrast of reconstructed tomograms was enhanced to improve the quality of automated and manual segmentation and for the visual quantification of filamentous tethers. We used IsoNet[74] for "missing wedge" correction and denoising: A content-representative fraction of our CTF-corrected 4-times binned tomograms was used to train an IsoNet model for 30 iterations of "missing wedge" restoration coupled with denoising by progressive introduction of Gaussian noise at iterations 10, 15, 20, 25 with the respective noise levels of 0.05, 0.1, 0.15, 0.2. The trained IsoNet model was then applied to restore the "missing wedge" and enhance contrast in all CTF-corrected 4-times binned tomograms.

**Segmentation.** Automated segmentations of synapses (Figs. 1 and 2g) were performed with MemBrain v2[75] on CTF-corrected, 4-times binned IsoNet-restored tomograms. Membranes (intracellular organelles and plasma membranes) were segmented automatically with MemBrain v2 and corrected manually using Amira (Thermo Fisher). The segmentations were re-colored and aligned with tomogram slices using ChimeraX v1.8. Manual segmentations of AZs (Fig. 4a) were made with IMOD v4.11 using 4-times binned, IsoNet-corrected tomograms.

**Morphometric characterization of membrane rearrangements.** All synapses were screened for membrane rearrangements potentially resembling fusion events. From these ROIs, 4-times binned sub-tomograms with a box size of 200 x 200 x 200 pixels (pixel size 12.28 Å) were generated using *dynamo_catalogue*[76]. As a quality control and to avoid bias, the original tomograms were screened, and ROIs were preselected by the first author. The second author double-checked all positions independently and generated the sub-tomograms. Initial ROIs were excluded if the putative fusion event was not located at an AZ with recognizable postsynaptic density, if a halo resembling a clathrin coat around the pore was visible, or if the resolution of the tomogram was poor (see Supplementary Fig. 6 for examples of excluded ROIs). The subtomograms were denoised and corrected for missing wedge effects using IsoNet. These denoised subtomograms were used for the classification of putative fusion states. In addition, we double-checked the correct classification for a subset of ROIs using the original (undenoised) dataset. Based on our observations, we defined 7 categories of membrane rearrangements at the AZ (opposing the postsynaptic density), also see Supplementary Fig. 7 for morphometric characteristics of each category.

Center slices of putative fusion events were exported from IMOD as tiff files. Morphometric measurements (Fig. 3a and Supplementary Fig. 7) were performed in fiji: the horizontal diameter of SVs with membrane invagination, stalk formation, closed, and open fusion pores (indicated as measurement a in Supplementary Fig. 7) was measured at the widest region of the respective SV/fusion event between the outer borders of the lipid bilayers. The width of dilating and collapsing fusion pores, as well as bumps, was measured between the two positions of the AZ membrane where inward curvature was observed (Supplementary Fig. 7, measurement a in bottom row). For the height of stalks, closed, open, dilated, and collapsing fusion pores, the distance between the AZ membrane outer (upper) border and the outer border of the fusion event membrane at the highest position was measured, whereby invaginations of the AZ membrane (stalk formation, closed fusion pore) were interpolated (Supplementary Fig. 7, measurement b). For the distance of tethered SVs and SVs during stalk formation (Supplementary Fig. 7, measurement c), the space between lipid bilayers of SV and membrane was measured, whereby membrane in- and evaginations were interpolated. The height of membrane invaginations (state 1; Supplementary Fig. 7, measurement d) was measured between the highest point of the invagination and the outer border of the AZ membrane. To discriminate between state 1 and state

2, the roundness of SVs was assessed by fitting a circle to the outer border of the SV. Distances between the circle and outer borders of the SV at the highest (Supplementary Fig. 7, measurement e1) and lowest (e2) position were measured. If the ratio of $e2 \times e1^{-1}$ was > 1.3, the SV shape was defined as a droplet (state 2). To discriminate between state 2 and state 3, the continuity of SV and AZ membrane was measured via grayscale intensity measurements along a vertical line scan spanning over 3 nm (horizontally) from the inner leaflet of the SV/fusion pore to the bottom border of the AZ membrane (Supplementary Fig. 7, measurement f). A "gap" (as indicator for missing continuity) between the SV and AZ membrane was defined as follows: mean grayscale intensity values and standard deviations (STD) were calculated for the line scan. Pixels with values below mean-STD were identified. If regions of at least 7 pixels, corresponding to 1 nm, with values below mean-STD were identified in the line scan, also local grayscale intensity maxima above and below this minimum were identified. For the minimum and both maxima, 7 pixels (1 nm) were averaged to account for noise, and the average of the two maxima regions was calculated. If the ratio of minimum-region × maxima-region$_{avg}^{-1}$ was < 0.6, the fusion event was defined as stalk formation containing a "gap" between SV and AZ membrane. If no minimum was detectable, we set minimum-region × maxima-region$_{avg}^{-1}$ = 1. This was the case for 4 out of 6 fusion events in category 3. The height of the neck of the closed fusion pore was measured between the inner leaflet of the bottom membrane of the fusion event and the bottom membrane of the AZ, meaning the region in which the side walls of the fusion pore were in direct contact (Supplementary Fig. 7, measurement g). To discriminate between state 3 and state 4, we measured if the pore neck of the fusion event was closed or open (Supplementary Fig. 7, measurement h). Analogous to measurement f, we measured if the pore neck was continuous (state 3) or contained a "gap" (state 4). Grayscale intensity values were thereby measured as horizontal line scans between outer borders of the pore neck at the narrowest position. Again, a gap was defined as minimum-region × maxima-region$_{avg}^{-1}$ < 0.6. We did not detect a minimum in any of the fusion events of category 3. The pore width of the open fusion pore was defined as the space between inner lipid bilayers of the pore walls at the narrowest position of the neck (Supplementary Fig. 7, measurement i). To discriminate between state 4 and state 5, we measured if outward curvature was present (j1, state 4) or not (j2, state 5). The angle j1 was measured between the vertical membrane of the pore neck and a line fitted to the bottom of the fusion pore head. To discriminate between state 5 and state 6, the thickness of membranes was measured at the top of the pore (Supplementary Fig. 7, measurement k1) and next to the pore base (k2). If the top of the fusion pore was thickened ($k1 \times k2^{-1} > 1.3$), it was defined as collapsing pore (state 6). The height ((Supplementary Fig. 7, measurement b) was used as a criterion to discriminate between state 6 and state 7, whereby collapsing fusion pores were defined as events with b > 10 nm.

**Quantification of fusion events, multivesicular release and bumps.** Based on our definition of membrane rearrangements, we quantified numbers of observations per category. We first quantified synapses containing at least one of these events (Fig. 3d). If synapses contained more than one event, only the event closest to category 4 (as center) was used to define the overall state. Since stimulated synapses contained more events of categories 2–6 than TTX-treated synapses, we defined these states as "ongoing SV fusion". Based on this definition, we had three groups of synapses: synapses with ongoing fusion, synapses with bump(s), and synapses without membrane rearrangements (Fig. 3d, e). Synapses containing at least two fusion events (categories 2-6) were counted as MVR synapses (Fig. 5c, d). In addition, we counted each fusion event individually (Fig. 3f).

**Quantification of SV distances, tethers and interconnectors.** The quantification of distances and filamentous tethers connecting SVs

and the AZ membrane (Fig. 4d–g), as well as distances between SVs/ fusion events and neighboring SVs (Fig. 5g–j) was performed in IMOD v4.11 using IsoNet-corrected tomograms. To make sure that the observed filaments were not a denoising artifact, we double-checked our annotations in a subset of ROIs using the original (un-denoised) dataset. For distance measurements, we first labeled all membrane-near SVs of each synapse above the AZ up to a distance of 24 nm at lower zoom and then measured their exact distances at higher zoom using the IMOD measuring tool. The distance between the SV and AZ membrane was measured at the center slice of the SV and defined as the space between the lipid bilayers of the SV and the membrane. We analyzed distributions of membrane-near SVs per synapse, whereby synapses containing no or only one SV were excluded. For this, we binned distances in 2 nm steps. Based on this distribution, previous reports[10,54] and our observation that membrane invaginations were visible below SVs with an average distance of 6.4 nm, we further defined three subpools of SVs: membrane-proximal SVs with a max. distance of 6 nm, intermediate SVs with a distance of 6–12 nm, and distal SVs with a distance of 12–24 nm.

Only synapses with very good structural resolution (high signal-to-noise ratio, clearly visible membrane bilayers, etc.) were used for tether analysis. Tethers were defined as vertical or angular filamentous connections between the SV and AZ membrane. We quantified tethers per SV manually, whereby we went back and forth in the z direction several times and at different zooms. The slicer window was additionally used to rotate SVs around the x and y axis. Only if a filament was visible on multiple z slices, it was counted. To reduce bias, synapses of both groups (non-releasing vs. releasing synapses) were analyzed in parallel.

For the analysis of distances to neighboring SVs, minimal distances between fusion events/ membrane-proximal SVs and nearby SVs were measured in x, y and z using the IMOD measuring tool. These distances were not necessarily measured from the center slice of the SV. If connecting filaments (interconnectors) between fusion events and neighboring SVs were observed in several consecutive slices and/ or via x/y rotation using the slicer window of IMOD, these fusion events were counted as interconnected (Fig. 5i, j).

**Simulation**
The coarse-grained model of the SV and the AZ includes particle-based representations of bilayer membranes, curvature-inducing proteins, and bead and spring models of tether and SNARE proteins. For the membranes, we used the membrane model developed by Sadeghi et al.[77]. This two-particle-per-thickness coarse-grained model is parameterized to mimic the mechanics of a fluid membrane with specified bending rigidity, has tunable in-plane viscosity, and is coupled with a hydrodynamics model that reproduces the out-of-plane kinetics of membranes in contact with solvents of prescribed viscosity[78,79].

To parameterize the membrane model, we used reported values of the elastic response of SVs to indentation forces in atomic force microscopy (AFM) measurements[80]. We used these values in conjunction with a theoretical model, initially developed for the AFM indentation of influenza virus envelopes[81], that relates the overall stiffness (or spring constant) of a spherical vesicle to the bending rigidity of its membrane. We found a mean membrane bending rigidity of $0.8 \times 10^{-19}$ J, which is well within the range of values obtained for lipid bilayers[82].

We modeled the SV in the initial state as a sphere with the outer diameter of 42.5 nm, to reflect the mean values obtained from tomograms (Fig. 3a), and added a harmonic volume-preserving potential to its outer leaflet particles. This potential acts against any changes in the enclosed volume, while allowing for otherwise arbitrary deformations. The plasma membrane underneath the vesicle is modeled as a planar square membrane patch of 180 nm in side length. The simulation box

is coupled in-plane to a stochastic barostat that controls the lateral pressure components around zero.

We modeled the curvature-inducing proteins in the AZ (Fig. 3g) via a force field masking mechanism that allows for tagged particles to locally modify the interparticle interactions using Monte Carlo moves, while letting these particles freely diffuse within the membrane. The modified force field reflects a preferred signed curvature (upward/ downward) around these particles[83]. We used reported values of membrane curvatures upon binding cyclic peptides derived from synaptotagmin-1 C2B domain to assign preferred local curvatures to these particles[84].

The tether proteins (Fig. 3g) are included as fixed-length elements initially formed between particles on the vesicle to anchor particles on the plasma membrane. The position of these tethers are decided randomly at the initial state. We incorporated a harmonic angle-bending potential that, when activated, exerts a torque to rotate the tethers about their anchor point on the plasma membrane, in effect pulling the SV toward the AZ.

We included chain-like representations of two sets of SNARE proteins, namely v-SNARE on the SV and t-SNARES in the AZ. The sizes of the chains roughly match the overall structure of SNARE proteins synaptobrevin, syntaxin, and SNAP-25 in the zippered complex. We included selective short-range attractive pairwise interactions between beads that form v-SNARE and t-SNARE chains such that they prefer to match one-to-one in the correct zippered configuration. Spatial exclusion, modeled via soft harmonic repulsions, energetically prohibits other conformations. We chose the copy number of SNARE proteins based on the reported proteomics data for SVs[47], and calibrated the strength of attractive interactions to match the data on zipping/ unzipping forces measured with magnetic tweezers[85].

To test the effect of curvature-inducing proteins in the AZ, we developed models with 0, 10, 20, and 30 copies of these proteins. For each model, we started 5 simulation replicas, each with different random distributions of tether, SNARE, and membrane-curving proteins. We used anisotropic Brownian dynamics with a timestep of 0.1 ns to simulate the motion of all the particles in the system, and assigned a cytosolic viscosity of 2.21 cP[86] to calculate particle mobilities in our hydrodynamic coupling method[79].

At the start of each simulation, the system is allowed to relax for 5 μs, with the torque on the anchor points of tether proteins disabled, effectively having the SV floating at a constant distance with the AZ (Supplementary Fig. 10a). Afterwards, the tethers are activated, pulling the vesicle toward the AZ. The gap between the two membranes in their closest approach is continuously monitored. When the gap falls below 6 nm, we initiate the force field masking mechanism for curvature-inducing proteins, which results in curvature being developed in the AZ membrane. Each simulation thus continues for another 100 μs to follow the docking dynamics (Supplementary Fig. 10).

Trajectories are obtained by sampling the positions of all the particles at 100 ns intervals. All the subsequent trajectory post-processing, data analysis, and plotting is done through Python scripts, using Numpy[87] and Matplotlib[88] software packages. The 3D visualization is done via the software package Visual Molecular Dynamics (VMD)[89].

### Statistics and data representation
Fluorescence microscopy data were analyzed using fiji v1.54 (fluorescence intensity means and histograms) and Python, cryo-ET data were quantified using IMOD and fiji. GraphPad Prism v8 was used for calculations of mean, standard error of the mean (sem), median, 95% confidence interval (CI) of the median, normality and statistical tests. Datasets were tested for Gaussian distribution using D'Agostino-Pearson, Shapiro-Wilk and Anderson-Darling tests. If datasets were normally distributed according to these tests, two-tailed unpaired t-tests (to compare two groups) were performed.

Otherwise, two-tailed Mann-Whitney tests (two groups) or Kruskal-Wallis tests (more than two groups) and Dunn's posthoc tests were performed, corrections for multiple comparisons were included. For paired, two-tailed comparison, a Wilcoxon test of pairs was performed. Two-tailed Fisher's exact tests for $2 \times 2$ tables were performed in GraphPad Prism, followed by Benjamini-Hochberg correction for multiple comparisons. For Fisher's exact tests of more complex tables, R v4.5 was used. P-values were defined as follows: * $p < 0.05$, ** $p < 0.01$, *** $p < 0.005$. Graphs were generated using the python packages Seaborn[90] and Matplotlib[88] and optimized for visualization using Affinity Designer 2. Schematic illustrations were generated with Affinity Designer v2 (Fig. 1b, c and Supplementary Fig. 7) and Microsoft 3D builder (Supplementary Fig. 3a). Isosurfaces of bandpass-filtered tomograms and segmentations were visualized using ChimeraX, whereby the "hide dust" function was used. Only for subtomogram averages, the erase tool was used to remove artifacts that were not in direct contact with the particle and introduced/ amplified by C61 symmetry. If not stated differently, data are represented as mean ± sem. In violin plots and bar graphs, the center line depicts the median, the upper and lower lines/borders of the box display the 25% and 75% percentiles. Whiskers in box plots indicate the 10–90% percentile range in all graphs except Supplementary Fig. 10 (here min. to max.). XY plots show lines connecting means, and the semitransparent areas indicate the sem.

### Ethical statement
All experimental procedures involving the use of mice were approved by the Animal Welfare Committee of the Charité-Universitätsmedizin Berlin and the Berlin State Government. For all experiments, 0–2 days old C57/BL6/N wildtype mice of either sex were used. Mice were housed in individually ventilated cages, room temperature: $22 \pm 2\,°C$, relative humidity: $55 \pm 10\%$, light/dark cycle of 12/12 h.

### Reporting summary
Further information on research design is available in the Nature Portfolio Reporting Summary linked to this article.

## Data availability
All data supporting the findings of this study are available within the paper and its Supplementary Information. The printing scheme for 3D pedestals used in cell culture is provided as Source Data. Source Data referring to graphs and statistical analyses of Figs. 2–5, as well as Supplementary Figs. 1, 4, 7, and 11, are provided with this paper. The following previously published PDB structures were used in this study: 5W5D, 4JEU, 7UDB, 7T7V. Source data are provided in this paper.

## Code availability
All codes developed for analyzing the simulation data, reproducing the corresponding figures and statistics, and constructing the Markov state models are publicly available at: [https://github.com/MohsenSadeghi/meso_synaptic_vesicle_fusion][91]. Simulation trajectories can be obtained from a public repository at https://ftp.mi.fu-berlin.de/pub/msadeghi/synaptic_vesicle/.

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

## Acknowledgements

We thank Heike Lerch, Berit Söhl-Kielczynski and members of the Rosenmund lab for technical assistance. We thank Marion Weber-Boyvat, Pascal Fenske, Melissa Herman and Severin Dicks for help with live imaging and the respective data analysis. We thank Metaxia Stavroulaki for help with cryo-confocal microscopy and CLEM, and Timo Flügel and Simon Lauer for help with plunge freezing. We thank Artsemi Yushkevich for help with data processing. We thank Melissa Herman and Erik Jorgensen for critical reading of the manuscript. We thank Thorsten

Trimbuch and the Viral Core Facility of the Charité-Universitätsmedizin Berlin for lentivirus and AAV design and production. Funding: Walter Benjamin Position from the DFG, project number 458275811 (to J.K.), postdoc fellowship of the DiGiTal program by the Berliner Chancengleichheitsprogramm (to J.K.), Reinhard Koselleck project, project number 399894546, Excellence cluster EXC2049 NeuroCure, project number 390688087, and NeuroNex project, project number 436260754, from the DFG (to C.R.), Kekulé fellowship from the Chemical Industry Fund of the German Chemical Industry Association (to U.K.), Heisenberg Award, project number 447835500, from the DFG (to M.K.). Young Investigator position at DFG collaborative research center (CRC) 1114 (to Mo.S.). Major Research Instrumentation from the DFG, project numbers 384148553 and 384149399, and BUA 512-ACEM WP1 from the Berlin University Alliance (to the Core Facility for Cryo-Electron Microscopy).

## Author contributions

J.K. and C.R. designed the study, C.R. supervised the project. J.K. and Ma.S. conceived the setup for optogenetic plunge freezing. CAD designed and built the optogenetic freezing device. J.K. developed a protocol to culture neurons on EM grids. J.K. and Ma.S. performed plunge freezings. J.K. and T.S. acquired cryo-ET data. U.K. and M.K. processed cryo-ET data. J.K. and U.K. analyzed cryo-ET data and segmented tomograms. Mo.S. generated the computational models and performed the corresponding simulations and analyses. J.K. acquired and analyzed cryo-confocal microscopy and CLEM data. L.I. and J.K. acquired live imaging and RT confocal microscopy data. J.K. and L.I. analyzed live imaging data. M.L. acquired and analyzed electrophysiology data. J.K. designed figures and prepared the manuscript. All authors reviewed the manuscript.

## Funding

## Competing interests

The authors declare no competing interests.
