## [Transparent Peer Review file · Nature Communications]

Dynamic nanoscale architecture of synaptic vesicle fusion in mouse hippocampal neurons

Corresponding Author: Dr Jana Kroll

Version 0:

Reviewer comments:

Reviewer #1

(Remarks to the Author)

The authors set up a time-resolved system to observe synaptic vesicles in situ. Optogenetics stimulation of action potential is achieved milliseconds before vitrification, and verified by monitoring fluorescence of a glutamate sensor. This is a nice set up, potentially useful to other applications. The authors collect cryo-tomograms of synapses and perform a thorough morphological analysis to derive fusion states, arrangement of tethers, etc. This analysis contributes to our understanding of synapses, for instance discriminating between current models for fusion. I have several points that require clarification.

1. The rationale for the double light pulse is unclear to the non-expert
2. When comparing fluorescence at 34C and in cryo, a less pronounced effect of stimulation is reported. This is attributed to the fact that bright neurons were selected in warm conditions, while in cryo the choice of cells to image was made blindly. In addition, CLEM was not used to target synapses for tomography. So we have to assume that a certain percentage of synapses that contribute to the cryo-ET dataset are not stimulated. Is it possible to infer what this percentage might be by comparison with the ration of bright/total neurons in warm conditions? How could this affect the analysis/conclusions?
3. Fig 3 a-d represents possibly the most important conclusion from this study, where the authors derive a temporal sequence of events within the milliseconds between stimulation and neurotransmitter release. Are those differences seen in panel d statistically significant? Dilated fusion pores are the same in stimulated and control cells: this is not mentioned in the text, but it requires mention and an explanation.
4. Related to point 3, the authors claim no evidence for tight docking. What if it was a very short-lived intermediate? Can the authors calculate on how long lived any intermediate must be to have any probability to be detected in 1/60 tomograms given their experimental set up?
5. Regarding the subtomogram averaging around the fusion point for the various states. I suppose C61 symmetry was chosen to average around the membrane-to-vesicle axis. (Why C61?) I couldn't understand from the methods whether this angle was determined by connecting the fusion point with the vesicle centres or if it was assumed to run perpendicular to the axis of view? (The latter case being potentially a significant approximation)
6. How are 'tethers' identified? Is it based on manual detection of densities such as that shown in 4j? Was this done in tomograms that were CTF corrected? Were they denoised? It seems to me this is by far the weakest part of the analysis. There is no evidence the density in 4j is more real than any of the other dark 'blobs' visible in the background. I would suggest removing any attempts to fit tethers and other molecules. I find this weakens the paper due to the obviously speculative nature, rather than adding to it.
7. The finding that clusters of vesicles form, and that some vesicles are seen attached to actin filaments: this is potentially interesting. But can the author calculate whether either of these two events is seen with a more than random frequency, given the number of vesicles and of actin filaments in the volumes analysed? If not, I'm afraid this part of the analysis is too speculative.

(Remarks on code availability)

Reviewer #2

(Remarks to the Author)

Reviewer Report for ESCI-2024-00928: "Dynamic nanoscale architecture of synaptic vesicle fusion in mouse hippocampal neurons"

Overall Summary:

This study presents a technically advanced integration of optogenetic stimulation, rapid cryofixation, and cryo-electron tomography (cryo-ET) to capture synaptic vesicle (SV) fusion intermediates in cultured mouse hippocampal neurons. The authors report observing synaptic activity using cryo-preserved iGluSnFR3 fluorescence and categorize diverse fusion intermediates, highlighting "stalk formation" as a predominant pre-fusion state. Key findings challenge the requirement for tight docking/hemifusion, demonstrate proximal SV depletion post-stimulation, and note filaments potentially linking fusion to vesicle resupply.

Major Strengths:

1. **Methodological Integration:** The combined optogenetics/cryo-freezing/iGluSnFR3 validation is a significant technical achievement, enabling millisecond-resolution snapshots of fusion dynamics.
2. **Scale of Analysis:** The dataset of 75 stimulated synapses provides substantial morphological information on fusion intermediates under near-native conditions.
3. **Insight into Fusion Mechanism:** The high frequency of "stalk formation" intermediates aligns with *in silico* models of local lipid splaying and challenges hemifusion-diaphragm models.

Major Concerns: Novelty of Workflow vs. Biological Insight:

1. The core strength lies in the biological findings enabled by the technique (stalk prevalence). Framing the workflow itself as the primary novelty requires greater distinction from existing rapid-freezing cryo-ET approaches (e.g., Radecke et al. EMBO Rep 2023; Fernandez-Busnadiego et al. JCB 2010, 2013; Papantoniou et al. Sci Adv 2023; or similar method such as 'zap-and-freeze' Kusick et al. Nat. Neurosci. 2020). Claims that this specific workflow yields "more near-native" states leading specifically to "more stalk formation" require significantly stronger support. Quantitative comparison of intermediate frequency (especially stalks) with datasets from prior cryo-ET studies using chemical stimulation or synaptosomes is absent. Is the observed stalk prevalence genuinely due to enhanced preservation, or influenced by differences in stimulation paradigm (optogenetic vs. chemical/electric), neuronal model (cultured neurons vs. synaptosomes), or analysis criteria? Rigorous comparison is crucial to attribute novelty to the workflow.
2. **Subjectivity in Fusion Intermediates Classification:** While Supplemental Table 1 provides valuable definitions for the 7 fusion states, the reliance on qualitative descriptors (e.g., "droplet-shaped," "slightly invaginated," "minimal dimple") introduces subjectivity, especially at state boundaries (e.g., State 1 "invagination" vs. State 2 "stalk"). The manuscript does not detail objective, quantitative metrics applied during classification (e.g., curvature measurements, distance/bilayer continuity thresholds derived from subtomogram features) or procedures to minimize bias (e.g., blinded classification). Without quantitative, unbiased classification rules applied consistently, the statistical significance of the reported distributions (Fig. 3d-f) and subsequent interpretations is diminished.
3. **Interpretation of State Distribution as Temporal Progression:** A minor thought related to the previous point. Could the relative abundance of states (Fig. 3f) reflect the duration of metastable intermediates? Which then implies state 2 (stalk) presents a higher kinetic barrier. However, several critical caveats challenge this interpretation:
 - Strictly Frozen Moment: The data captures vesicles frozen at various states all within a narrow temporal window (2-5 ms post-stimulation). This represents only one specific "slice" in time after stimulation onset. It does not capture the full temporal evolution of individual vesicles from initial contact to collapse (which would require serial capture across multiple timepoints).
 - Assumption of Linear Progression: Interpreting the distribution as reflecting progression times assumes all vesicles traverse the defined states in sequence. This model remains hypothetical.
 - Potential Sampling Bias: The observed distribution is also influenced by capture efficiency (volatility, detectability) and stochastic variation inherent in snapshots.
 - Unknown Kinetics at Other Timepoints: A very different distribution might be observed at slightly earlier (e.g., 1 ms) or later (e.g., 10 ms) timepoints post-stimulation. For instance, vesicles destined for rapid fusion might bypass several intermediates entirely or be underrepresented in the 2-5 ms window.

Specific Points Requiring Clarification / Enhancement:

1. The authors repeatedly emphasize that their method is "time-resolved". However, they didn't utilize their instruments, which indeed has the capability to do time-lapsed freezing, to generate measurements at different time points of the optogenetics stimulation.
2. The description in line 269-271 is not accurate. If fraction of category 7 for stimulated and TTX-treated groups are alike, what about stage 5?
3. On page 11, second paragraph, the copy number range of curvature-inducing proteins shall be addressed before mentioning 0 or 10 copies. Otherwise this is very confusing.
4. The description on line 333 is too vague.
5. On page 13, second paragraph, the authors seem to define a criterion and soon afterwards use it to validate their workflow? (line 365-367)

Conclusion and Recommendations: This study presents a valuable methodological advance and rich cryo-ET dataset capturing SV fusion intermediates in a near-physiological context. The key biological finding is the high prevalence of "stalk formation" as the major fusion initiation pathway under these spatiotemporal conditions. However, the novelty of the workflow relative to extant methods needs clearer delineation, the objectivity of fusion state classification requires strengthening, and the interpretation of intermediate abundance as a direct proxy for kinetic dwell times must be significantly moderated.

Recommendation: Major Revision

(Remarks on code availability)

Version 1:

Reviewer comments:

Reviewer #1

(Remarks to the Author)

The authors responded satisfactorily to almost all my comments.

I would like to challenge them on point 7: if the difference in filaments is not significant, then their presence next to vesicles is just as likely to be by chance than not. I therefore do not think this observation should be reported, or at least not in the current form.

If reported, the authors need to be very explicit about the fact that it is impossible to know whether any specific interactions are present and that further analysis is needed.

(Remarks on code availability)

Reviewer #2

(Remarks to the Author)

The authors have made substantial revision to the manuscript, "Dynamic nanoscale architecture of synaptic vesicle fusion in mouse hippocampal neurons." In this version, the authors have made reasonable alterations to address some of the more difficult questions raised by the reviewers. The current manuscript provides crucial contribution to the field, and matches the standard of publication in Nature Communications. I recommend the publication of this manuscript.

(Remarks on code availability)

Point-by-point response

We sincerely thank both reviewers for their valuable feedback, which helped us to further improve the quality, clarity, and comprehensibility of our manuscript. Our revised manuscript now contains quantitative criteria for the classification of vesicle fusion states (Fig. 3 and Suppl. Fig. 7) and a nearest neighbor analysis for fusion events and vesicles (Fig. 5g-j). We generated Markov State Models for the progression of vesicle fusion with and without tight docking (Suppl. Fig. 9). We extended our discussion: we now highlight technical advances of our workflow compared to existing workflows and added a paragraph about the limitations of our study.

Reviewer #1 (Remarks to the Author):

1. The rationale for the double light pulse is unclear to the non-expert

We added explanations to the results section, lines 119-121.

2. When comparing fluorescence at 34C and in cryo, a less pronounced effect of stimulation is reported. This is attributed to the fact that bright neurons were selected in warm conditions, while in cryo the choice of cells to image was made blindly. In addition, CLEM was not used to target synapses for tomography. So we have to assume that a certain percentage of synapses that contribute to the cryo-ET dataset are not stimulated. Is it possible to infer what this percentage might be by comparison with the ration of bright/total neurons in warm conditions? How could this affect the analysis/conclusions?

We calculated which fractions of recordings were excluded during data analysis due to changes in fluorescence intensity lower than 10%. For the glutamate sensor iGluSnFR3.v857.GPI, which was used for cryo-confocal microscopy, we excluded 15/66 recordings (22.7%) because of $\Delta F/F_0 < 0.1$. In 5 of these recordings, we were still able to fit rise and decay functions, meaning that we were not able to identify changes in fluorescence intensity in 13.6% of all recordings with this glutamate sensor. We chose the threshold of 0.1 because the primary aim of the recordings was the characterization of on- and off-kinetics. We now report the exclusion rate in the results section (lines 168-171), Supplementary methods and Source table 15.

A direct comparison of these live imaging results with our cryo-ET experiments is challenging because of the different experimental setups: for live imaging, we applied electrical field stimulation. Although this method allowed us to directly induce bouton depolarization, the electrical field is typically not homogeneous throughout the imaging chamber but stronger near the electrodes and weaker in the center, where most recordings are performed due to the optics of the setup. This means that although the depolarization is more direct, we cannot guarantee that action potentials were reliably induced in all performed recordings. On the other hand, we were able to induce single action potentials in 90-100% of synapses using optogenetics. When applying two light pulses (40Hz frequency) ~80% of synapses responded to both pulses (Suppl. Fig. 1d). In these cells, we were not able to optically measure neurotransmitter release due to the overlapping wavelengths of ChR2 activation and iGluSnFR3 excitation. Consequently, we do not have a readout for fractions of induced action potentials in the live imaging data, whereas we do not have a readout for fractions of releasing neurons following action potential induction.

However, if we assume a synaptic release probability of ~85%, as recently reported for hippocampal neurons using optical measurements (Dürst et al. Nat Com 2022, DOI:

10.1038/s41467-022-33565-6), it is likely that in nearly all neurons action potentials were induced in our live imaging setup. Considering that optogenetic stimulation led to action potentials in 80-90% of all neurons, we may expect fusion events in 68-76.5% of all synapses in our cryo-ET samples. Indeed, the fraction of synapses without membrane rearrangements in the group of stimulated synapses was 33.3%, which closely matches this calculation. Yet, the number of synapses with ongoing fusion (according to our definition) was lower than 68-76.5%, indicating that we either captured a fraction of ~20% synapses postfusion or that our optogenetic cryo-ET workflow led to a lower synaptic release probability, e.g. due to the culturing system on grids.

In the previous version of our manuscript, we utilized the synaptic release probability of ~85% to calculate whether the distribution of membrane-proximal SVs in the fraction of stimulated synapses without ongoing fusion resembles more likely non-releasing or postfusion synapses. We added additional information about our live imaging experiments to this section (lines 328-332).

3. Fig 3 a-d represents possibly the most important conclusion from this study, where the authors derive a temporal sequence of events within the milliseconds between stimulation and neurotransmitter release. Are those differences seen in panel d statistically significant? Dilated fusion pores are the same in stimulated and control cells: this is not mentioned in the text, but it requires mention and an explanation.

We performed statistical tests for panels 3d and 3e and now mention them in the respective figure legend and results section (lines 232-236 and 243-244). All results can be found in Source Table 4. We mention that numbers of dilated fusion pores are comparable in the results (line 234) and comment on this observation in the discussion (lines 523-531).

4. Related to point 3, the authors claim no evidence for tight docking. What if it was a very short-lived intermediate? Can the authors calculate on how long lived any intermediate must be to have any probability to be detected in 1/60 tomograms given their experimental set up?

Due to the limited temporal readout of our setup, we were not able to calculate exactly how long-lived an intermediate must be to be detected. However, to estimate if a (very short-lived) tight docking step could be included in the fusion sequence suggested by us, we generated Markov state models without and with tight docking (Suppl. Fig. 9). These models provide relative waiting times for transitions between states. According to these models, a tight docking step prior to or following stalk formation is very unlikely, as described in the results section (lines 269-274).

5. Regarding the subtomogram averaging around the fusion point for the various states. I suppose C61 symmetry was chosen to average around the membrane-to-vesicle axis. (Why C61?) I couldn't understand from the methods whether this angle was determined by connecting the fusion point with the vesicle centres or if it was assumed to run perpendicular to the axis of view? (The latter case being potentially a significant approximation)

We revised our Supplementary method section and added explanations for the symmetry axis and the C61 symmetry.

6. How are 'tethers' identified? Is it based on manual detection of densities such as that shown in 4j? Was this done in tomograms that were CTF corrected? Were they denoised? It seems to me this is by far the weakest part of the analysis. There is no evidence the density in 4j is more real than any of the other dark 'blobs' visible in the background. I would suggest removing

any attempts to fit tethers and other molecules. I find this weakens the paper due to the obviously speculative nature, rather than adding to it.

We performed the tether (between SV and active zone membrane) and interconnector (between SVs) analysis in tomograms that were CTF-corrected by phase flipping using IMOD, followed by denoising and “missing wedge” correction using IsoNet. We now describe these post-processing steps in more detail in the Methods section “Contrast enhancement” (lines 685-693). To ensure consistency of the visually identified features before and after IsoNet-correction, we double-checked our results in a fraction (at least 20%) of tomograms again in CTF-corrected but undenoised (non-IsoNet-corrected) datasets. Although the analysis of tethers is prone to bias, it is still one of the most common cell biological electron microscopy measurements to detect subtle changes in the organization of SV pools. Automated quantification tools for tether/interconnector analysis (e.g., “Pyto package”, used here: Papantoniou et al. Sci Adv 2023 (DOI: 10.1126/sciadv.adf6222) or here: Radecke et al. EMBO Rep 2023 (DOI: 10.15252/embr.202255719)) and manual analyses (e.g., Michanski et al. EMBO Rep 2023 (DOI: 10.15252/embr.202256702)) are likewise accepted in the field. The manual analysis of tethers in our study was performed by the first author, who is an expert in electron microscopy image analysis with ~10 years of hands-on experience. Generally, the tether analysis was only performed in the tomograms with the best signal-to-noise ratio (in total 34 synapses of stimulated neurons). To distinguish tethers from noise (“dark blobs”), several consecutive tomogram slices were screened: only structures that appeared in several slices were counted as tethers. Additionally, the slicer window of IMOD was used to rotate the SV in x and y direction and to generate averaged views. To reduce bias, the analyzer used subtomograms (box size 200x200x200 pixels) and analyzed synapses of both groups (non-releasing vs. releasing synapses) in parallel.

We agree with the reviewer that our data do not provide the necessary structural information to unambiguously fit atomic models to isosurfaces of tethers. We revised the respective Suppl. Fig. (now Suppl. Fig. 12) and now only position atomic models next to tomogram slices. We further show more examples of tethered SVs.

We added a “limitations of our study” section to the discussion. We comment on the subjective nature of tether analysis there (lines 531-538).

7. The finding that clusters of vesicles form, and that some vesicles are seen attached to actin filaments: this is potentially interesting. But can the author calculate whether either of these two events is seen with a more than random frequency, given the number of vesicles and of actin filaments in the volumes analysed? If not, I’m afraid this part of the analysis is too speculative.

We performed analyses of nearest neighbor distances between fusion events and close-by SVs. For comparison, we also quantified nearest neighbor distances for membrane-proximal SVs. The results are now shown in Fig. 5g-j. We further analyzed presynaptic cytoskeletal filaments in more detail. Since we did not observe significant differences in the organization of presynaptic filaments between releasing and non-releasing synapses, we now show the respective tomogram slices and a quantification of filament distances to the active zone in Suppl. Fig. 13. Although not significantly different, we believe that the identification of connections between putative actin filaments and SVs is of high enough interest for the neuroscience community to keep it as part of this manuscript.

Reviewer #2 (Remarks to the Author):

1. The core strength lies in the biological findings enabled by the technique (stalk prevalence). Framing the workflow itself as the primary novelty requires greater distinction from existing rapid-freezing cryo-ET approaches (e.g., Radecke et al. EMBO Rep 2023; Fernandez-Busnadiego et al. JCB 2010, 2013; Papantoniou et al. Sci Adv 2023; or similar method such as 'zap-and-freeze' Kusick et al. Nat. Neurosci. 2020).

Claims that this specific workflow yields "more near-native" states leading specifically to "more stalk formation" require significantly stronger support. Quantitative comparison of intermediate frequency (especially stalks) with datasets from prior cryo-ET studies using chemical stimulation or synaptosomes is absent.

Is the observed stalk prevalence genuinely due to enhanced preservation, or influenced by differences in stimulation paradigm (optogenetic vs. chemical/electric), neuronal model (cultured neurons vs. synaptosomes), or analysis criteria? Rigorous comparison is crucial to attribute novelty to the workflow.

We performed a thorough comparison of several studies combining stimulation and electron microscopy. We provide the resulting table here with our point-by-point response and added the most important points to our discussion (lines 426-452).

Several factors influenced why we observed a higher prevalence of stalk formation compared to previous studies. Most strikingly, the overall number of observed fusion events in our study is higher than in any other cryo-ET study thus far. Beyond that, the timing in our optogenetic workflow was ideally suited to capture SV fusion events, whereas earlier studies used chemical stimulation applied for minutes (Fernández-Busnadiego et al. JCB 2010 (DOI: 10.1083/jcb.200908082)) or a spraying technique in which earliest time points were reported as ~7 ms post-stimulation (Radecke et al. EMBO Rep 2023 (DOI: 10.15252/embr.202255719)). Although the temporal resolution (meaning variance in action potential induction) in electrically stimulated samples is typically even lower compared to optogenetics, recent studies combining electrical stimulation and electron microscopy failed to capture stalk formation. The reason here is that the thereby used freeze substitution (e.g. Kusick et al. Nat Neurosci 2020 (DOI: 10.1038/s41593-020-00716-1)) leads to a lower spatial resolution. On the one hand, the space between SV stalk and active zone membrane of less than 5 nm is very likely masked in TEM projections from 40 nm thick sections of resin-embedded samples. On the other hand, freeze-substituted samples typically contain heavy metal stains, which are incorporated into cellular structures for contrast enhancement.

	Cryo-ET studies				Room temperature EM studies	
Publication	This study	Fernández-Busnadiego et al. JCB 2010	Radecke et al. EMBO Reports 2023	Glynn et al. Cell Rep Methods 2025	Watanabe et al. Nature 2013	Kusick et al. Nat Neurosci 2020
Sample	Mouse hippocampal neurons cultured on EM grids (banker culture)	Rat cerebrocortical synaptosomes	Rat synaptosomes	Mouse brain tissue (cortex or hippocampus)	Mouse hippocampal neurons cultured on sapphire glass disks (astrocyte co-culture)	Mouse hippocampal neurons cultured on sapphire glass disks (astrocyte co-culture)
Freezing device	Plunge freezing (Vitrobot)	Plunge freezing (device not mentioned)	Plunge freezing (home-built plunge freezer)	High-pressure freezer (Leica EM ICE)	High-pressure freezer (Leica HPM 100)	High-pressure freezer (Leica EM ICE)
Freezing temperature	37°C	Not mentioned	23–25°C	Room temperature	34°C	37°C
Freezing buffer, key components	4 mM Ca ²⁺ and 1 mM Mg ²⁺ (to increase release probability); 3 μM NBQX and 30 μM bicuculline to block network activity	1 mM MgCl ₂ , calcium not mentioned; 30 mM KCl for chemical depolarization	1.3 mM CaCl ₂ (~physiological conc.) and 1 mM MgCl ₂ in buffer; 1 mM CaCl ₂ and 52 mM KCl in spraying solution	Combinations of phosphate buffer/ PBS and cryoprotectants, including Dextran, sucrose, ethylene glycol	4 mM Ca ²⁺ and 1 mM Mg ²⁺ (to increase release probability); 3 μM NBQX and 30 μM bicuculline	1.2 mM CaCl ₂ and 3.8 mM MgCl ₂ (both ~physiological conc.); additional exp. with elevated CaCl ₂ 3 μM NBQX and 30 μM bicuculline
Stimulation	Optogenetic, 5 ms light pulse to induce one action potential	Chemical, 30 mM KCl for 1 min	Chemical, 52 mM KCl sprayed on grid during plunging	No stimulation	Optogenetic, 10 ms light pulse to induce one action potential	Electrical, 10 V·cm ⁻¹ applied for 1 ms across a 6 mm space
Fusion initiation	Via action potential generation (mediated by sodium influx)	Via direct membrane depolarization and opening of VGCCs	Via direct membrane depolarization and opening of VGCCs	spontaneous	Via action potential generation (mediated by sodium influx)	Likely via action potential generation (direct membrane depolarization possible, too)

Temporal resolution	Vitrification on average 7-8 ms after light onset, 1-5 ms after action potential generation (mainly 2-3 ms)	None, vesicle fusion not timed and network activity not blocked within 1 min stimulation	Vitrification 7 ms or 35 ms after spraying, stated temporal resolution "few milliseconds or even sub-milliseconds"	none	Vitrification between 6 and 12 ms after action potential generation	Vitrification earliest 5 ms after action potential generation, ± 1 ms
Confirmation of stimulation	Direct, glutamate sensor for cryo-confocal microscopy	none	Direct, via diffusion marker	none	none	Indirect, via FM 1-43FX uptake assay live before freezing
Sample treatment post-freezing	None	None	None	Focused ion beam milling (lift-out)	Freeze substitution using glutaraldehyde and osmium tetroxide, resin embedding, sectioning (40-70 nm for 2D, 200 nm for tomography)	
Microscope		CM300 (Philips) and T30 Polara (FEI), operated at 300 kV, field emission gun, 2k \times 2k charge-coupled device camera (Gatan), post-GIF energy filter (Gatan)	Tecnai F20, operated at 200 kV, field emission gun, 2k \times 2k CCD camera (Gatan); and Titan Krios, K2 direct electron detector (Gatan) without energy filter	Titan Krios G4 operated at 300 kV, Selectris Energy Filter, Falcon 4i direct electron detector camera	Zeiss 902, Megaview III camera; and Tecnai F20, 4k CCD camera (FEI Eagle) for tomography	Phillips CM 120, operated at 80 kV on the $\times 93,000$ setting, AMT XR80 camera
Pixel/voxel size		0.68 (CM300) and 0.66 nm (Polara); binned twice during reconstruction, final voxel size 2.72 and 2.64 nm	0.75 or 1.2 nm (Tecnai) and 0.6 nm (Krios)	0.31 or 0.2 nm	Not mentioned	Not mentioned

N analyzed tomograms/ synapses/ micrographs		Untreated 7 tomograms; 30 mM KCl 6 tomograms	Control 9 tomograms; stimulated 9 tomograms (6 late, 3 early)	56 tomograms containing 107 synapses acquired,	Unstimulated 193 micrographs, stimulated 193 micrographs; additional 43 synapses for tomography	Unstimulated 274 micrographs, stimulated 315 micrographs (at least 2 samples); additional serial sections
Definition and numbers of fusion events		 • Vesicles with open pore n = 2 • Patches of AZ of high, concave curvature n = 2 (full-collapse fusion events) 	Early-stage events:  • Vesicle and plasma membrane slightly bent toward each other, no contact n = 8 • Contact between vesicle and plasma membrane bilayer n = 3 • Point contact n = 5 Late-fusion stage events:  • Small SV fusion pore opening n = 3 • Wide fusion pore opening n = 1 • Small bump n = 11 	Fusion events in 7% of synapses Observed fusion states ranging from initial SV tethering, stalk/fusion pore formation, fusion pore opening and collapse	Fusing vesicles: “omega figures” Random 2D sections: fusion pits in 19% of micrographs from stimulated samples, 1% in unstimulated	Definition exocytic pits: smooth curvature at AZ in an otherwise straight membrane (greater than 10 nm), not mirrored by postsynapse Random 2D sections: exocytic pits in 18% of micrographs from stimulated samples, 2% in unstimulated; Serial sections: exocytic pits in 35% of stimulated synapses, 3% in unstimulated

Observation of stalk formation (as defined in our study)	Yes	Not mentioned	Yes	One example, either stalk or closed fusion pore	No (technically challenging in >40 nm sections)	No (technically challenging in >40 nm sections)
Observation of tight docking (membranes in direct contact)	Yes, n = 1	No	No	Not mentioned	Yes	Yes

Additional electron tomography studies using optogenetics for synapse stimulation in acute brain slices (Borges-Merjane 2020 <https://www.sciencedirect.com/science/article/pii/S0896627319310888?via%3Dihub>, no fusion events) and Imig 2020: Omega profiles described, various size (mossy fiber)

2. Subjectivity in Fusion Intermediates Classification: While Supplemental Table 1 provides valuable definitions for the 7 fusion states, the reliance on qualitative descriptors (e.g., "droplet-shaped," "slightly invaginated," "minimal dimple") introduces subjectivity, especially at state boundaries (e.g., State 1 "invagination" vs. State 2 "stalk"). The manuscript does not detail objective, quantitative metrics applied during classification (e.g., curvature measurements, distance/bilayer continuity thresholds derived from subtomogram features) or procedures to minimize bias (e.g., blinded classification). Without quantitative, unbiased classification rules applied consistently, the statistical significance of the reported distributions (Fig. 3d-f) and subsequent interpretations is diminished.

We thank the reviewer for raising this important point. We defined quantitative classification rules for all described states and now describe them in Suppl. Fig. 7 via schematic illustrations of all measurements. We extended our method section and now also explain the measurements and calculations here in more detail (lines 716-766). We updated Fig. 3 and added missing values to Fig. 3a. We now summarize all measurements in source table 16.

3. Interpretation of State Distribution as Temporal Progression: A minor thought related to the previous point. Could the relative abundance of states (Fig. 3f) reflect the duration of metastable intermediates? Which then implies state 2 (stalk) presents a higher kinetic barrier. However, several critical caveats challenge this interpretation:

Strictly Frozen Moment: The data captures vesicles frozen at various states all within a narrow temporal window (2-5 ms post-stimulation). This represents only one specific "slice" in time after stimulation onset. It does not capture the full temporal evolution of individual vesicles from initial contact to collapse (which would require serial capture across multiple timepoints).

Assumption of Linear Progression: Interpreting the distribution as reflecting progression times assumes all vesicles traverse the defined states in sequence. This model remains hypothetical.

Potential Sampling Bias: The observed distribution is also influenced by capture efficiency (volatility, detectability) and stochastic variation inherent in snapshots.

Unknown Kinetics at Other Timepoints: A very different distribution might be observed at slightly earlier (e.g., 1 ms) or later (e.g., 10 ms) timepoints post-stimulation. For instance, vesicles destined for rapid fusion might bypass several intermediates entirely or be underrepresented in the 2-5 ms window.

We agree with the reviewer that not only the transition kinetics between states but also other factors likely influence the numbers of observed fusion events per state. As suggested by the reviewer, acquiring a time series would help supporting our assumption of linear progression based on the shapes of the fusion intermediates. This would require a huge additional effort in data acquisition, that we like to pursue in future studies. We fully agree with the reviewer that the temporal progression suggested by the order of the structures shown in Figure 3 is not tested experimentally. While we feel that the order of structures as displayed are the most logical one, given what we know when vesicles undergo collapse during fusion, the assumed order is not strictly relevant for our conclusions. Important is that we observed at the time point of membrane arrest several different structural intermediates that are absent in unstimulated synapses, and this argues that in the few ms after release the structural changes of the collapsing vesicle are highly dynamic. To provide some feasibility check for the order, we nevertheless generated Markov State Models reflecting relative transition times between states (in response to reviewer 1), which are based on numbers of observed events. We make

clear that these models are simplified and do not account for a putative sampling bias or several reversible steps during fusion. We discuss this point within the “limitations of our study” section in the discussion (lines 538-552).

Specific Points Requiring Clarification / Enhancement:

1. The authors repeatedly emphasize that their method is “time-resolved”. However, they didn’t utilize their instruments, which indeed has the capability to do time-lapsed freezing, to generate measurements at different time points of the optogenetics stimulation.

We changed “time-resolved” to “timed”, where appropriate. We comment in the discussion that our workflow has the potential for millisecond temporal resolution (lines 419-424) but that we studied a single time point/ time span (lines 541-543).

2. The description in line 269-271 is not accurate. If fraction of category 7 for stimulated and TTX-treated groups are alike, what about stage 5?

We revised the respective results section (line 236) and comment on the discrepancy in the discussion (lines 523-531).

3. On page 11, second paragraph, the copy number range of curvature-inducing proteins shall be addressed before mentioning 0 or 10 copies. Otherwise this is very confusing.

Done (lines 287-291).

4. The description on line 333 is too vague.

We specified this sentence (lines 311-312).

5. On page 13, second paragraph, the authors seem to define a criterion and soon afterwards use it to validate their workflow? (line 365-367)

We clarified that the criteria (distributions of SVs under different conditions) are based on previous studies (lines 323-324).

Point-by-point response

Reviewer #1 (Remarks to the Author):

The authors responded satisfactorily to almost all my comments.

I would like to challenge them on point 7: if the difference in filaments is not significant, then their presence next to vesicles is just as likely to be by chance than not. I therefore do not think this observation should be reported, or at least not in the current form.

If reported, the authors need to be very explicit about the fact that it is impossible to know whether any specific interactions are present and that further analysis is needed.

We thank the reviewer for the positive feedback.

We agree that this point needs further investigation. Following the reviewer's recommendation, we removed the supplementary figure and the corresponding Results section.

Reviewer #2 (Remarks to the Author):

The authors have made substantial revision to the manuscript, "Dynamic nanoscale architecture of synaptic vesicle fusion in mouse hippocampal neurons." In this version, the authors have made reasonable alterations to address some of the more difficult questions raised by the reviewers. The current manuscript provides crucial contribution to the field, and matches the standard of publication in Nature Communications. I recommend the publication of this manuscript.

We thank the reviewer for the positive feedback.